

**The cooperative application of oyster shell and biochar efficiently enhanced**
**in-situ remediation of cadmium contaminated soil around intensive industry**
Bin Wu[a*], Jia Li[a], Mingping Sheng[b], He Peng[b], Dinghua Peng[b], Heng Xu[b*1]
[a] State Environmental Protection Key Laboratory of Synergetic Control and Joint Remediation for
Soil & Water Pollution, College of Ecology and Environment, Chengdu University of Technology,
Chengdu, 610059, PR China
[b] Key Laboratory of Bio-Resource and Eco-Environment of Ministry of Education, College of
Life Sciences, Sichuan University, Chengdu, 610065, PR China

[1]  Corresponding author: E-mail address: wub@cdut.edu.cn (Bin Wu); xuheng64@sina.com (Heng Xu)



## Abstract

Biochar has been widely used for the in-situ remediation in the cadmium (Cd)
contaminated soil, while the high-cost of biochar limited its application in farmland.
In this study, we firstly investigated the possibility of cooperative application of
oyster shell and biochar to enhance Cd immobilization efficiency and reduce the cost
in field experiments under rice-oilseed rape rotation. Treatments were comprised of:
rice planting without amendments (R-PA0); followed with 15000 kg/ha biochar
(R-PA1); followed with 15000 kg/ha oyster shell (R-PA2); followed with 7500 kg/ha
biochar and 7500 kg/ha oyster shell (R-PA3); rice-oilseed rape rotation without
amendments (RT-PA0); rotation with 15000 kg/ha biochar (RT-PA1); rotation with
15000 kg/ha oyster shell (RT-PA2); rotation with 7500 kg/ha biochar and 7500 kg/ha
oyster shell (RT-PA3). Results revealed that HOAc-extractable Cd was significantly
decreased by 38.46% in R-PA2. Cd contents in brown rice and oilseed were reduced
by 29.67% in R-PA3 and 19.74% in RT-PA3 compared with control. Meanwhile,
Hazard Quotient of brown rice and oilseed significantly decreased in RT-PA3. The
Olsen-P in R-PA3 and RT-PA3 was markedly increased by 187.46% and 184.73%,
respectively. In addition, activities of urease, catalase, and β-galactosidase in RT-PA3
were significantly increased by 268.88%, 30.44% and 245.28%, respectively.
Furthermore, the jiont application of biochar with oyster shell significantly decreased
the cost of soil remediation at least 9600 RMB/ha. These results demonstrated that the
joint utilization of biochar with oyster shell might be an economical and effective
pathway to achieve in-situ remediation of Cd contaminated farmland.





**Keywords:** Biochar; Oyster shell; Rice-oilseed rape rotation; In-situ remediation;
Enzyme activities; Cadmium

## 1. Introduction

Heavy metals have been considered as hazardous materials for human health,
and among which cadmium (Cd) is one of the most toxic heavy metals (Yang et al.,
2021). The excessive intake of Cd could cause serious damages to bones, thyroid, and
kidneys (Ma et al., 2021a). According to the latest national survey on the status of soil
environmental quality in China, Cd has ranked as the highest contaminants (7%)
among all heavy metals (Mou et al., 2020). In southwest China, a large amount of
farmlands were contaminated with Cd owing to the intensive industrialization (Chen
et al., 2018a). In addition, soil acidification also aggravated the bioavailability and
solubility of Cd, thus enhancing Cd uptake by crops (Feng et al., 2020). Therefore, the
development of cost-effective and eco-friendly remediation technologies is crucial for
food safety and soil quality.
In recent years, in-situ immobilization as an effective technology has raised wide
attentions in the remediation of Cd contaminated farmlands, which can reduce the Cd
uptake by plants without delaying agricultural production (Palansooriya et al., 2020;
Wang et al., 2021). In recent years, biochar derived from bio-wastes is widely
recommended as a soil amendment (Zong et al., 2021). Amounts of nutrients (such as
C, N, P, K, and Mg etc.) in biochar could improve soil fertility and promote plant
growth (Lu et al., 2015). Moreover, biochar has a large surface area and plenty of
functional groups, which are reactive to immobilize heavy metals, including Cd, Pb,





and Ni (Wang et al., 2021). However, the high price of biochar limited its large
application. In addition, the soil pH regulation by biochar in acid fields was not very
significant, while the soil pH values were negatively related to Cd availability (Liu et
al., 2018). Therefore, it is vital to decrease the remediation cost of biochar without
reducing immobilization efficiency of Cd. Oyster shell is a low-cost and largely
available bio-waste product from oyster farming (Li et al., 2020). Previous studies
found that oyster shell as a promising slow-release alkaloid has outstanding effects on
pH adjustment and Cd immobilization in soils (Chen et al., 2018b; Peng et al., 2020).
In this sense, we think that the joint use of biochar and oyster shell might be a
low-cost and effective pathway to decrease Cd uptake by crops and improve soil
biochemical quality in acidic fields. However, there was little known about the joint
effects of biochar and oyster shell on the in-situ remediation in Cd contaminated soil.
Rice and oilseed rape were the main food and economic crops in southwest
China, and rice-oilseed rape rotation is the dominant production model (Liu et al.,
2014). Previous studies mainly focused on the effects of amendments on reducing the
Cd uptake by rice, while the remediation efficiency of passivators under the rice-rape
rotation was little known. Based on above opinions, a filed experiment under
rice-oilseed rape rotation was designed: (1) to investigate the cooperative effects of
biochar and oyster shell on Cd immobilization; (2) to evaluate the effects of biochar
and oyster shell on decreasing human health risk of Cd; (3) to reveal the effects of
biochar and oyster shell on soil biochemical properties including pH, CEC, total
organic carbon, organic matter, Olsen-P, Olsen-K, Alkeline-N, and the activities of





soil enzymes, so as to estimate the pearson correlation analysis (PCA) model of main
parameters in the moderately polluted farmland.
**2. Material and methods**
**2.1. Experimental site and soil properties**
A field trial was conducted during 2019-2020 in a rice-oilseed rape rotation
cultivated site where the soil was moderately contaminated by Cd. The field site was
located in a dominant agricultural cultivation region round industrial parks in
Chengdu plain, Sichuan province, China (104°18′N, 31°81′E). This region belongs to
a subtropical monsoon humid climate with an average temperature of 16.1 ℃ and
annual rainfall of about 1000 mm. The main properties of the topsoil (0 - 20 cm)
collected from the site in 2019 and 2020 were shown in Table S1.
**2.2. Characteristics of experimental materials**
Biochar was purchased from Zhenjiang Zedi agricultural and biological Co., Ltd.,
which was produced from rice straw in a reactor with $N_2$ and 500 ℃ for about 4 h.
Oyster shell was purchased from Fujian Mata Co., Ltd (< 0.3 mm mesh). The surface
structures of biochar and oyster shell were analyzed by Scanning Electron Microscope
(SEM, JSM-7500F). The functional groups of biochar and oyster shell were measured
by Fourier Transform Infrared Spectra (Nicolet 6700). The seeds of rice "Yixiang
2115" and seeds of oilseed rape "Yiyou 15" were obtained from Rice Research
Institute, Sichuan Academy of Agricultural Science.
**2.3. Experimental setup**
The field experiment was conducted during 2019 - 2020 as following treatments:





➢ R-PA0: Rice planting without passivators;
➢ R-PA1: Rice planting with 15 t/ha biochar;
➢ R-PA2: Rice planting with 15 t/ha oyster shell;
➢ R-PA3: Rice planting with 7.5 t/ha biochar and 7.5 t/ha oyster shell;
➢ RT-PA0: Rice-oilseed rape rotation without passivators;
➢ RT-PA1: Rice-oilseed rape rotation with 15 t/ha biochar;
➢ RT-PA2: Rice-oilseed rape rotation with 15 t/h oyster shell;
➢ RT-PA3: Rice-oilseed rape rotation with 7.5 t/ha biochar and 7.5 t/ha oyster
shell.
The concentrations of biochar and oyster shell used in this study were referred to
previous reports (Ameloot et al., 2014). Each experimental plot was 56 $m^2$ (7 x 8 m)
and arranged in a randomized design with three replicates. Before rice planting, the
passivators were sufficiently mixed with topsoil. After the harvest of rice, the oilseed
rape was planted following the conventional tillage pattern without any passivator.
**2.4. Plant analysis**
The rice grain and rapeseed were dried and ground to powder. Then, 0.2 g
samples were digested with $HNO_3$:$HClO_4$:HF in a mixture of 5:4:3 (v/v) and the
mixture was then diluted to 10 mL with 1% $HNO_3$ (Wu et al., 2019b). The Cd
concentrations in the mixture were determined by AAS.
**2.5. Soil analysis**
Soil pH was determined by a pH meter (METTLER-S220) with a soil/water ratio
of 5 g/25 mL. The bioavailable Cd of soil was measured by the TCLP method (Xu et





al., 2020). Briefly, 2 g of soil sample was mixed with 40 mL 0.11 M acetic acid and
shaken at 25 °C, 150 rpm for 16 h. Then, the mixture was centrifuged for 5 min at
8000 rpm and the supernatant was collected to determine Cd content by atomic
absorption spectroscopy (AAS; VARIAN, SpecterAA-220Fs). Olsen-P, Olsen-K, and
Alkeline-N were measured according to the method ascribed by (Liu et al., 2017).
Soil TOC and OM were determined by the method ascribed by (Walz et al., 2017).
In addition, activities of soil enzyme were analyzed to reflect the biological
quality in this study. Dehydrogenase activity was evaluated by the production of
triphenylfornazan (TPF) at $OD_{492nm}$ and expressed as μg TPF/g soil/24 h (Benefield et
al., 1977). Acid phosphate activity was assayed by the *p*-nitrophenol (pNP) release at
$OD_{400nm}$ and expressed as μg pNP/g soil/24 h (van Aarle and Plassard, 2010). Urease
activity was determined by the $NH_4$-complex at $OD_{578nm}$ and expressed as μg
$NH_4$-N/g soil/24 h (Yan et al., 2013). Catalase activity was measured by back titration
of $H_2O_2$ added to soil with 0.1 M $KMnO_4$ and expressed as by mL 0.1 M $KMnO_4$/g
soil/h (Zhang et al., 2011). Invertase activity was assayed by the amount of glucose
production at $OD_{508nm}$ and expressed as μg glucose/g soil/24 h (Wu et al., 2019b).
β-galactosidase activity was measured by the released 4-methylumbelliferone (MUF)
and expressed as μg MUF μmol/g soil/h (Martínez-Iñigo et al., 2009).
**2.6. Human risk assessment of Cd**
The health risks of Cd to adults and children in the crops were separately
assessed by the Hazard Quotient (HQ) according to the method introduced by
Environmental Protection Agency (EPA) in the US (Wei et al., 2020). HQ values were



calculated as the following formula:

HQ = (*EF* x *ED* x *C* x *IR*)/(*BW* x *AT* x *RfD*)         (1)

*EF* (Exposure Frequency): 365 days/year.

*ED* (Exposure Duration): 70 years for adult, 7 years for children.

*C*: Cd concentrations in the rice grain and oilseed (mg/kg).

*IR* (Ingestion Rate): For rice grain, 0.3892 kg/day for adult and 0.1984 kg/day for

children, respectively. For rape oil, 0.025 kg/day for adult and 0.0125 kg/days for
children, respectively.

*BW* (Body Weight): 62.71 kg for adult male, 55.1 kg for adult female and 25.6 kg

for children.

*AT* (Averaging Time): 25550 days for adult and 2555 days for children.

*RfD* (Reference of Dose): 0.001 mg/kg for Cd.

**2.7. Statistical analysis**

In this study, statistical significance was analyzed using SPSS 18.0 package, and

means values were considered to be different when $P < 0.05$ using least significant
difference (LSD). All statistics were performed using Origin 8.0 (USA).
**3. Results and discussion**
**3.1. Characteristics of soil and amendments**

The main characteristics of soil, biochar and oyster shell were shown in Table S1.

The soil in the field was acidic soil with pH values of 5.27 - 5.51. The biochar and
oyster shell used in the field study were alkaline materials and their pH values were
8.22 and 8.52, respectively. The OM of biochar (541.53 mg/kg) was significantly





higher than that of soil (39.32 mg/kg) and oyster shell (12.60 mg/kg). The carbon
percentage of biochar also reached 92.50%.
The surface of oyster shell (Figure 1a) was a regular filamentary layer with some
disordered deposition, which might be calcium compounds. The structure of biochar
(Figure 1b) was lamellar and polyporous, which was in favor of Cd absorption. In
addition, FTIR was operated to detect functional groups of oyster shell and biochar
(Figure 1c). The characteristic peaks of calcium carbonate in oyster shell were
observed at 1427 $cm^{-1}$ and 879 $cm^{-1}$ (Lu et al., 2021). Biochar showed obvious peaks
at 1089 $cm^{-1}$ and 790 $cm^{-1}$, which were related to C-O, and C-H bending vibration,
respectively (Wu et al., 2019a). In addition, an obvious feature at 3436 $cm^{-1}$
corresponding to -OH was loaded on oyster shell and biochar (Lian et al., 2021).
**3.2. Analysis of soil Cd bioavailability**
To evaluate the effect of different amendments on Cd bioavailability, the
concentrations of AcOH-extractable Cd in soils were determined by TCLP method
(Halim, 2003). Figure 2 showed the variations of AcOH-extractable Cd with different
amendments in rice-oilseed rape rotation. Compared to rice planting, the
concentrations of AcOH-extractable Cd increased in the oilseed rape planting, which
was mainly related to the different irrigation methods. In this study, rice planting was
performed by flooding irrigation during the whole physiological period whereas
dry-land cultivation was used in oilseed rape planting. In general, flooding irrigation
is beneficial to reduce Cd bioavailability due to the precipitation of Cd compounds
under low-redox status (Eh < 0 mV) (Mou et al., 2020). In addition, treatments with





biochar and oyster shell resulted in the reduction of AcOH-extractable Cd in soils.
Compared to R-PA0, the AcOH-extractable Cd was significantly decreased by
20.79% and 40.59% in R-PA1 and R-PA2, respectively. Compared to RT-PA0, the
AcOH-extractable Cd was also reduced by 5.76% and 17.85% in RT-PA1 and RT-PA2,
respectively. Moreover, the Cd immobilization efficiency in R-PA3 and RT-PA3 was
higher than that in R-PA1 and RT-PA1. These results demonstrated that oyster shell
gave a better Cd immobilization than biochar and the addition of oyster shell could
strength the Cd immobilization capacity of biochar.
**3.3. Analysis of Cd contents in brown rice and oilseed**
As shown in Figure 3, the application of biochar and oyster shell significantly
reduced Cd contents in brown rice and oilseed. Treatments without amendments, the
contents of Cd in brown rice reached 0.88 mg/kg. Compared to control (R-PA0), the
Cd contents in brown rice was decreased by 20.88% in R-PA1 and 30.77% in R-PA2,
indicating that oyster shell has the superior of Cd immobilization capacity than
biochar. Obviously, the cooperative addition of biochar and oyster shell (R-PA3)
contributed to higher Cd reduction (29.67%) in brown rice than that in sole biochar
(R-PA1, 20.88%). In addition, Cd contents in oilseed were significantly reduced in the
RT-PA1 and RT-PA3, about 27.63% and 19.74% lower than that in RT-PA0,
respectively. The results indicated that biochar and oyster shell application could
efficiently decrease Cd accumulation in brown rice and oilseed.
**3.4. Health risk assessment of cadmium**
HQ values of Cd for brown rice and oilseed intake in different treatments were





presented in Figure 4. The order of HQ for consuming rice and oilseed was children >
adult female > adult male, indicating that children were more sensitive than adults
under Cd exposure. Without amendments (R-PA0), HQ values of consuming brown
rice for adult male, adult female and children reached 5.46, 6.21 and 6.82,
respectively. After the application of amendments, HQ values of brown rice intake
were significantly decreased by 20.87%, 31.11% and 29.76% in R-PA1, R-PA2 and
R-PA3, respectively. Although HQ values of oilseed were significantly lower than that
of rice grain, but the values also decreased by 17.27 - 28.14% in the oyster shell and
biochar treatments.
**3.5. Analysis of soil biochemical properties**
**3.5.1. Analysis of soil pH and CEC**
It was observed that soil pH was weakly increased by biochar, but significantly
increased by oyster shell (Figure 2a). After the oyster shell application, the soil pH
increased to neutral (6.9 - 7.3) from acidity (5.2 - 5.5). Compared with control
(R-PA0), the soil pH was increased by 1.8, 1.6, 1.4 and 1.7 pionts in R-PA2, R-PA3,
RT-PA2 and RT-PA3, respectively. The application of oyster shell slight increased the
CEC in the rice planting, while oyster shell and biochar had no significant effects on
CEC in the rice-oilseed rotation (Figure 2b).
**3.5.2. Analysis of soil nutrients**
It is important for in-situ remediation of Cd contaminated soil by bio-wastes
without inhibiting the soil available nutrients. To analyze the effects of amendments
on soil bioavailable nutrients, the contents of TOC, OM, Olsen-P, Olsen-K, and



Alkeline-N were determined during the rice-oilseed rape rotation (Table S2). Biochar
application slightly increased TOC and OM in rice planting and rice-oilseed rotation.
In rice planting, TOC and OM in R-PA3 were increased by 10.09% and 9.92%
compared with R-PA0, respectively. In rice-oilseed rape rotation, soil TOC in RT-PA1
was enhanced by 11.06% and 11.32% compared with RT-PA0, respectively. More
obviously, Olsen-P was significantly increased by the addition of oyster shell.
Compared with R-PA0, the Olsen-P in R-PA2 and R-PA3 significantly increased by
200.96% and 187.46%, respectively. Compared with RT-PA0, the Olsen-P in RT-PA2
and RT-PA3 significantly increased by 295.92% and 184.73%, respectively.
**3.5.3. Analysis of soil enzyme activities**
As shown in Figure 6, adding amendments variously changed the activities of
soil enzyme. In the rice planting, biochar application increased the dehydrogenase
activity, about 20.12% (R-PA1) and 25.49% (RT-PA1) higher than that of control
(R-PA0). However, oyster shell significantly increased the dehydrogenase activity in
rice-oilseed rotation, which was markedly increased by 59.75% and 53.39% in the
RT-PA2 and RT-PA3 compared with control, respectively. Urease activity was no
obvious variation in the biochar treatment whereas markedly enhanced in the oyster
shell treatment. Compared with RT-PA0, urease activity was significantly increased
by 268.88% in RT-PA3. However, oyster shell and biochar had no obvious impacts on
acid phosphate activity, except for a reduction of 43.30% in R-PA2. In addition, the
application of biochar has no negative effects on invertase activity, while oyster shell
slightly decreased the invertase activity on rice-oilseed rape rotation. In the RT-PA3





treatment, catalase activity was significantly increased by the application of biochar
and oyster shell. Moreover, β-galactosidase activity was significantly increased by
245.28% in RT-PA0 with the maximum of 12.29 μg MUF μmol/g soil/h.

**3.6. Analysis of correlation coefficient**

To analyze and confirm the relationship among different parameters, the Pearson

correlation analysis was used to experimental data. As shown in Figure 7a, Cd content
in brown rice was positively correlated to Cd bioavailability ($R^2$ = 0.90) but
negatively correlated to soil pH ($R^2$ = -0.83). Meanwhile, the activities of soil
enzymes except acid phosphate were positively connected to alkaline-N, Olsen-P,
Olsen-K, and TOC. Figure 7b showed a weak correlation between Cd uptake of rape
and Cd bioavailability. In addition, soil pH was positively correlated to Olsen-P and
β-galactosidase activity ($R^2$ > 0.95), which further demonstrated that alkaline
substances could increase Olsen-P content and β-galactosidase activity by adjusting
soil pH in acid fields.

**3.7. Cost approach for amendments**

Considering the large scale remediation of the contaminated agricultural soil, the

cost of amendments is a key parameter in the practical application. The market price
of biochar ( > 1200 RMB/t) was much higher than that of oyster shell ( 500 RMB/t)
(detailed information see Supplementary Information). In this study, the dosage of
amendments was 12000 kg/ha. The market price for biochar amendment was at least
14400 RMB/ha, while the jiont use of biochar and oyster shell significantly decreased
the cost of amendments at least 9600 RMB/h. Furthermore, the jiont application of



biochar and oyster shell is more effective to improve soil biochemical properties
compared to biochar. Based on these consideration, the collaborative passivation of
biochar and oyster shell might be a economical pathway for the safe-use of Cd
contaminated soil.

**4. Discussion**

Rice and oilseed rape were the most important crops over the globe.
Simultaneously, rice and oilseed rape rotation was the main cultivated model in China.
However, the Cd contamination in agricultural lands, especially in acidic soils, has
severely threatened food safety production and human health. Cd accumulation in the
plants poses a gereat human health risk. Cd uptake by crops may result in kidney
damage and adverse effects on lung, cardiovascular, musculoskeletal systems (Wei et
al., 2020).
In-situ immobilization was an effective pathway to decrease Cd uptake by crops
by the application of amendments. Biochar is originated from bio-wastes, such as
straw, coconut shell and animal manure. Previous studies has revealed that biochar
has a great potential on Cd immobilization by surface absorption and co-precipitation.
However, the high price of biochar and weak Cd binding in acidic soil was limited its
large application in agricultural lands. In this study, our results showed that the
cooperative application of oyster shell and biochar could contribute to the reduction of
AcOH-extractable Cd in soils (Figure 2) and Cd uptake by crops (Figure 3).
Furthermore, the non-cancer risk description methodology of HQ was widely applied
to assess the possibility of health risk of Cd in different plants (Ma et al., 2021b). The



decreased HQ values demonstrated the human health risk of Cd decreased by the
application of oyster shell and biochar.

AcOH-extractable Cd has widely used to evaluate Cd bioavailability in soils, and

the reduction of Cd bioavailability was main mechanism of in-situ immobilization
(Liu et al., 2021). Soil pH was the main factor influencing the Cd bioavailability in
soils. It has been widely verified that soil pH determined solid-solution equilibria of
heavy metals in soils (Zhao and Masaihiko, 2007). Comparatively, soil pH in the
oyster shell treatment was higher that in biochar treatment (Figure 5). Although the
pH values of oyster shell (8.52) and biochar (8.22) were similar, oyster shell was
regarded as a low-release alkaloid in soils due to it is composed of CaO and $CaCO_3$.
The dissolution of CaO and $CaCO_3$ from oyster shell in water could produce hydroxyl
ion ($OH^-$) as the following chemical reactions (Ok et al., 2010):
$CaO + H_2O \rightarrow Ca^{2+} + 2OH^-$           (2)
$CaCO_3 + H_2O \rightarrow Ca^{2+} + HCO_3^- + OH^-$     (3)
An increase in soil pH can cause an increase in the negative soil surface charge, which
easily causes an increased capacity of cationic metal adsorption (Ok et al., 2010). The
precipitants of metal oxy/hydroxides could be formed due to increased hydroxyl ions
(Bolan et al., 2014). In addition, previous studies also found that functional groups
such as -OH and C-O loaded on the surface of oyster shell and biochar can decrease
the Cd solubility by surface adsorption and precipitation (Ok et al., 2010; Tang et al.,
2020). Therefore, the cation exchange, surface complexation and co-precipitation
might be mechanisms for the Cd immobilization of biochar and oyster shell in acidic



filed.
Olsen-P, Olsen-K and Alkeline-N play an important role on soil biochemical
quality and plant growth. P fractions are mainly dependent on soil pH, soil mineralogy
and phosphate fertilizer application (Lee et al., 2008). Fe-P and Al-P are the
predominant forms in acidic soils while calcium bound-P is the predominant form in
alkaline soils (Dean, 1949). In acidic soils, the loosely bound phosphates are
converted into Fe-P and Al-P fractions gradually owing to the re-precipitation process.
Previous studies found that Olsen-P content reaches the maximum at neutral pH soils
(Lee et al., 2008). Our results showed that the addition of oyster shell markedly
increased the content of Olsen-P in soils, which might be resulted from the
enhancement of soil pH (Table S2). PCA analysis (Figure 7) further demonstrated that
Olsen-P was highly correlated with changes of soil pH ($R^2 > 0.99$). The contents of
Olsen-K and Alkeline-N also slightly increased with the application of biochar and
oyster shell, indicating an improvement of soil fertility.
Activities of soil enzyme have been widely used to reflect soil biological quality
(Lin et al., 2021). In this study, activities of dehydrogenase, urease, acid phosphate,
invertase, catalase and β-galactosidase were determined, and most of which were
increased by the application of biochar and oyster shell (Figure 6). Especially, the
increase of activities of dehydrogenase, urease, catalase and β-galactosidase was
obvious under the stimulation of biochar and oyster shell. The increase of soil enzyme
activities might be explained from the following aspects. The addition of oyster shell
increased the soil pH, which usually caused the enhancement of dehydrogenase and



urease activities (Wen et al., 2021). Abd El-Azeem et al reported that dehydrogenase
activity was positively correlated with soil pH (Abd El-Azeem et al., 2013). Oyster
shell could raise the urease activity, thus catalyzing the hydrolysis of urea to $CO_2$ and
$NH_3$ with an optimum pH around 7.4 (Lee et al., 2008). In addition, the porous
structure and rich nutrients of biochar and oyster shell can contribute to the growth of
soil microorganisms, and thus might increase the soil enzyme activities (Azadi and
Raiesi, 2021; Wu et al., 2019a). Moreover, the enhancement of enzyme activities in
biochar and oyster shell treatments could also be related to the decrease of Cd toxicity
in soils (Zhang et al., 2021). In conclusion, the cooperative application of biochar and
oyster shell was the most effective pathway in the in-situ remediation of Cd
contaminated farmlands.
**5. Conclusions**

The current study revealed the effects of oyster shell and biochar on Cd

bioavailability, Cd uptake by crops, and human health risk of Cd as well as soil
biochemical properties during rice-oilseed rape rotation. The application of oyster
shell showed an extraordinary potential to increase soil pH for a duration, which
significantly decreased the Cd bioavailability in soils. The cooperative application
significantly reduced Cd contents and human health risk of brown rice and oilseed. In
addition, biochar application increased OM and TOC, while the addition of oyster
shell was suitable to improve Olsen-P, Olsen-K, and Alkeline-N. Furthermore, the
activities of soil enzyme were markedly enhanced by the cooperative application of
oyster shell and biochar. In addtion, Our results suggested that the joint application of





biochar and cheap oyster shell was a low-cost pathway to effectively reduce Cd
uptake of crops and improve soil biochemical properties.
**Acknowledgements**
This study was financially supported by Key Technologies Research and
Development Program (CN) (2018YFC1802605), the Science and Technology Project
of Sichuan Province (2022ZDYF0281) and Chengdu Science and Technology Project
(2021-YF05-00195-SN). The authors also wish to thank Professor Guanglei Cheng
and Hui Wang from Sichuan University for the technical assistance.

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

various pyrolysis temperatures: Cadmium immobilization mechanisms and
environmental implication. Bioresource Technology. 321, 124459.















**Figure captions:**
**Figure 1** SEM images of oyster shell (a) and biochar (b) and FTIR spectra (c) of
oyster shell and biochar.
**Figure 2** The effects of passivators on Cd bioavailability in soil. Dots represent the
value of each sample. Bars followed with different lowercase letters (a - c) and capital
letters (A, B) indicated significant ($p \leq 0.05$) difference among different treatments in
rice planting and oilseed rape planting according to the LSD test. Values represent
means ± standard deviation.
**Figure 3** The effects of passivators on Cd contents in brown rice (a) and oilseed (b).
Dots represent the value of each sample. Bars with different lowercase letters
indicated significant ($p < 0.05$) difference among different treatments according to the
LSD test. Values represent means ± standard deviation.
**Figure 4** The effects of different passivators on the HQ of grown rice and oilseed.
Mean with different lowercase letter indicated significant ($p < 0.05$) difference from
each other according to the LSD test. Values represent means ± standard deviation.
**Figure 5** The effects of different passivators on soil pH (a) and CEC (b). Dots
represent the value of each sample. Bars followed with different lowercase letters (a -
c) and capital letters (A, B) indicated significant ($p \leq 0.05$) difference among different
treatments in rice planting and oilseed rape planting according to the LSD test. Values
represent means ± standard deviation.
**Figure 6** The effects of different passivators on the activities of soil enzyme. Dots
represent the value of each sample. Bars followed with different lowercase letters (a -
c) and capital letters (A-C) indicated significant ($p \leq 0.05$) difference among different
treatments in rice planting and oilseed rape planting according to the LSD test. Values
represent means ± standard deviation.
**Figure 7** The correlation of investigated parameters in rice planting (a) and
rice-oilseed rape rotation (b)



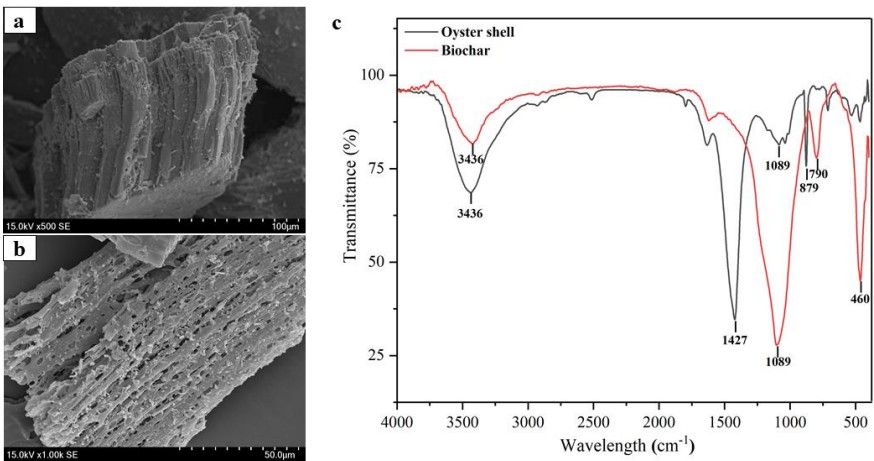

**Figure 1** SEM images of oyster shell (a) and biochar (b) and FTIR spectra (c) of oyster shell and biochar.

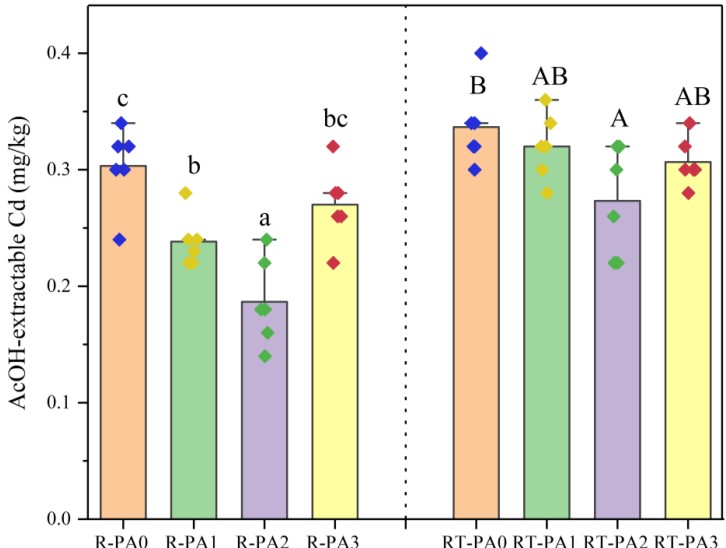

**Figure 2** The effects of passivators on Cd bioavailability in soil. Dots represent the value of each sample. Bars followed with different lowercase letters (a - c) and capital letters (A, B) indicated significant ($p \leq 0.05$) difference among different treatments in rice planting and oilseed rape planting according to the LSD test. Values represent means ± standard deviation.



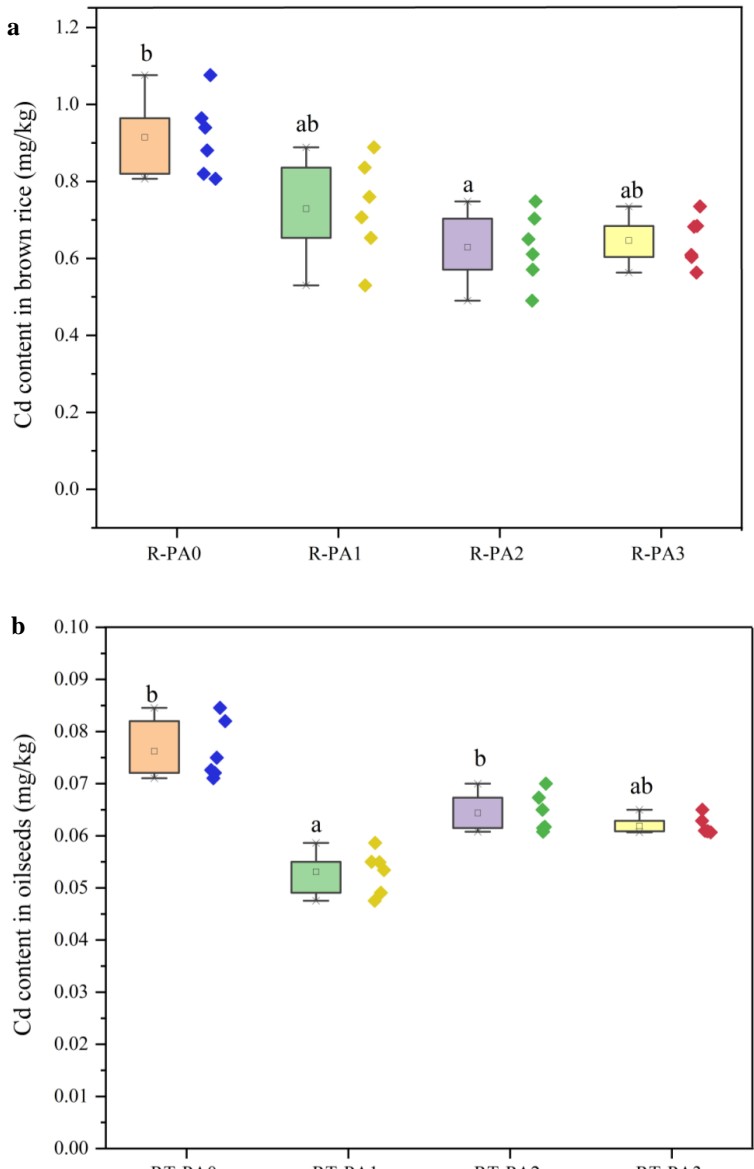

**Figure 3** The effects of passivators on Cd contents in brown rice (a) and oilseed (b).

Dots represent the value of each sample. Bars with different lowercase letters

indicated significant ($p < 0.05$) difference among different treatments according to the

LSD test. Values represent means ± standard deviation.



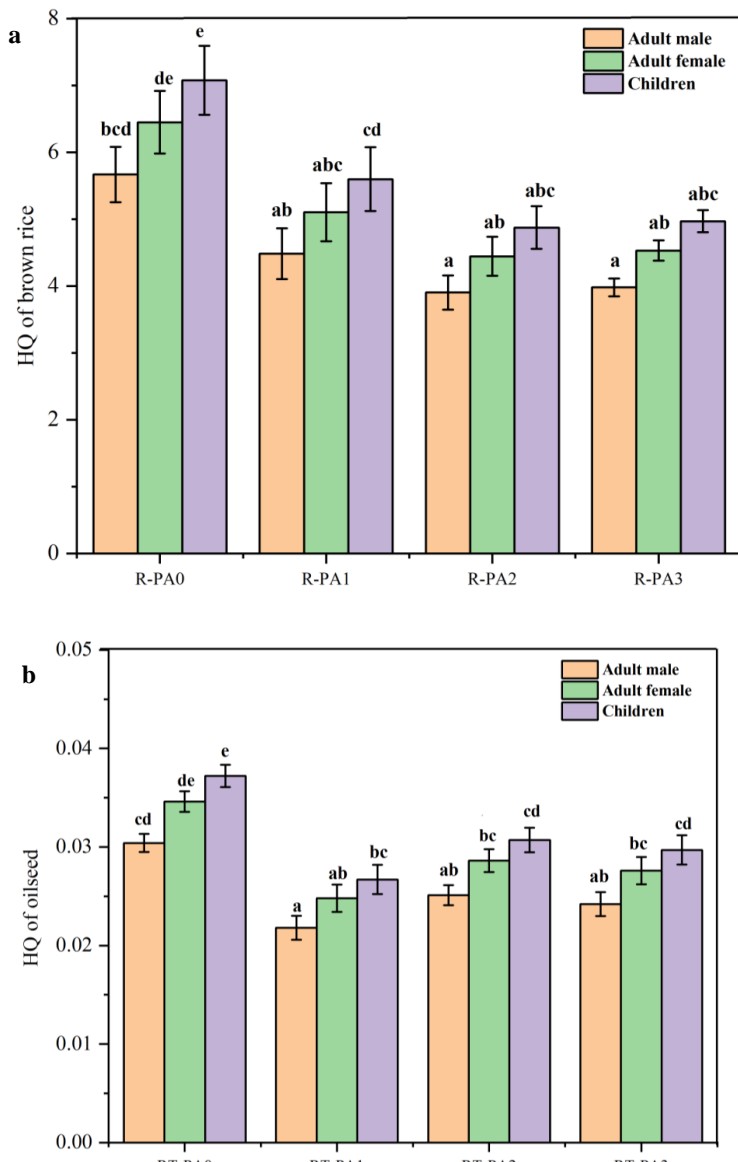

**Figure 4** The effects of different passivators on the HQ of grown rice and oilseed. Mean with different lowercase letter indicated significant ($p < 0.05$) difference from each other according to the LSD test. Values represent means ± standard deviation.



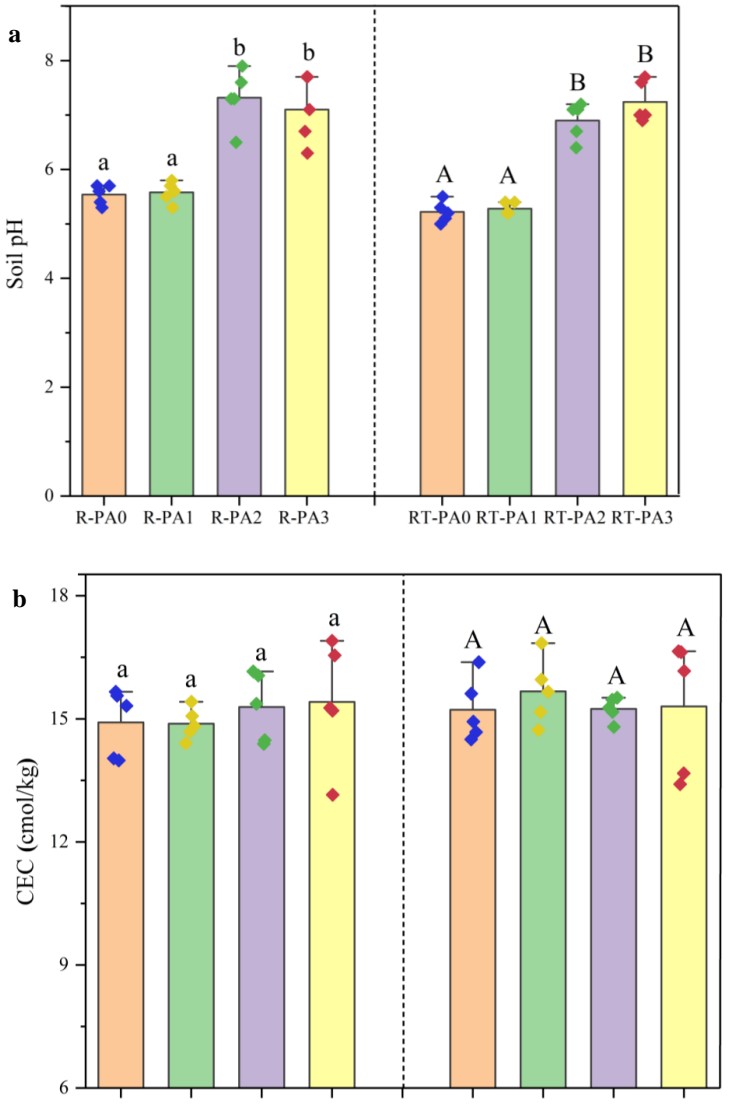

**Figure 5** The effects of different passivators on soil pH (a) and CEC (b). Dots represent the value of each sample. Bars followed with different lowercase letters (a - c) and capital letters (A, B) indicated significant ($p \leq 0.05$) difference among different treatments in rice planting and oilseed rape planting according to the LSD test. Values represent means ± standard deviation.



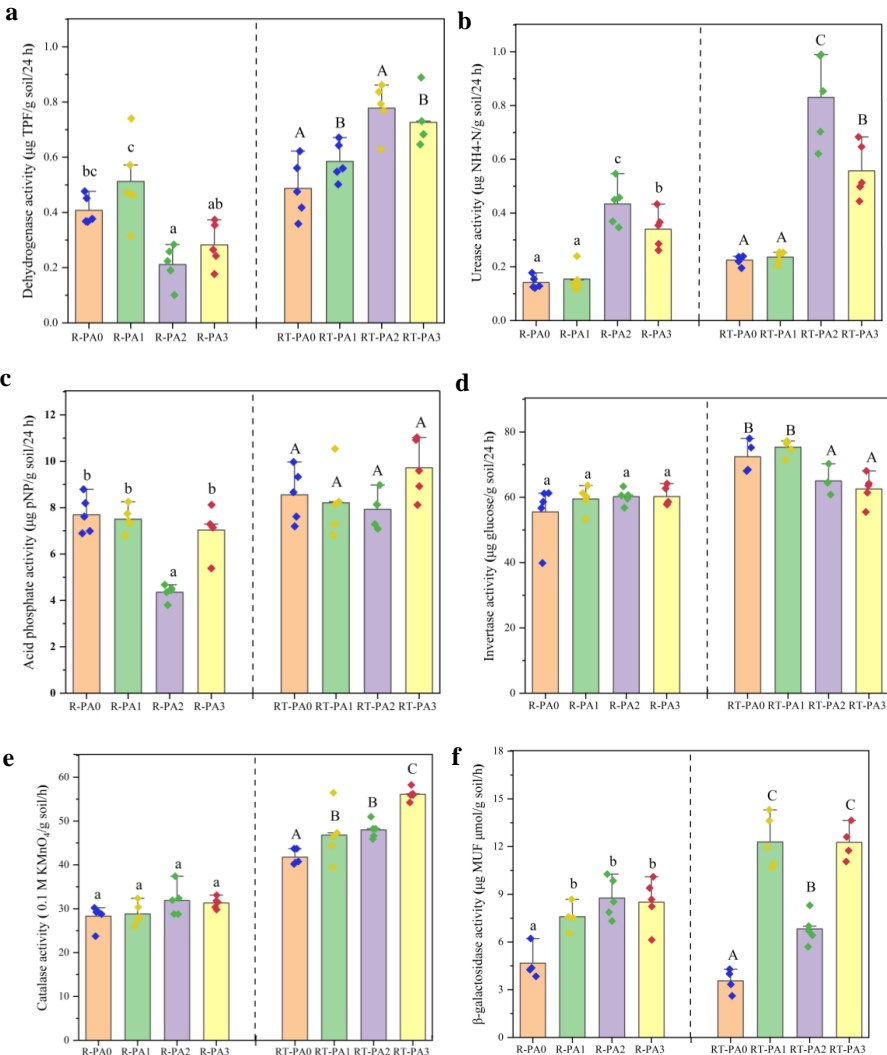

**Figure 6** The effects of different passivators on the activities of soil enzyme. Dots represent the value of each sample. Bars followed with different lowercase letters (a - c) and capital letters (A-C) indicated significant ($p \leq 0.05$) difference among different treatments in rice planting and oilseed rape planting according to the LSD test. Values represent means ± standard deviation.





**Figure 7** The correlation of investigated parameters in rice planting (a) and rice-oilseed rape rotation (b).



## Code/Data availability

The data refering to this paper was all presented in the Supplemental file.

## Author contribution

**Bin Wu:** Investigation, Writing Original Draft, Supervision

**Jia Li:** Writing - Review & Editing

**Mingping Sheng:** Investigation

**He Peng:** Investigation, Visualization

**Dinghua Peng:** Investigation, Data Curation

**Heng Xu:** Conceptualization, Resources, Funding acquisition

## Declaration of interests

The authors declare that they have no known competing financial interests or personal relationships that could have appeared to influence the work reported in this paper.