# Peer review of "The cooperative application of oyster shell and biochar efficiently enhanced in-situ remediation of cadmium contaminated soil around intensive industry"

_SOIL, 2021_

## Author Comment (AC1)

**Soil**

**Manuscript No.:** SOIL-2021-145

**Manuscript title:** The cooperative application of oyster shell and biochar efficiently enhanced in-situ remediation of cadmium contaminated soil around intensive industry

**Article type:** Research paper

**Authors:** Bin Wu, Jia Li, Mingping Sheng, He Peng, Dinghua Peng, Heng Xu

**Reviewer 1:**

The authors of the manuscript Soil-2021-145 "The cooperative application of oyster shell and biochar efficiently enhanced in-situ remediation of cadmium contaminated soil around intensive industry" addressed an interesting and important topic by using amendments (biochar and oyster shell) to achieve the safe-production of crops in the Cd contaminated soil following the rice-oilseed rape rotation. Indicators reflected soil quality including soil fertility and enzyme activities were also investigated by authors. Cd contaminant in farmland has been causing a great concern on human health. Overall, the authors present a well-designed study with appropriate methods, and this study provided a practical method to reduce the Cd contents in crops and reduce the remediation cost. I think that this study is well-worth to be published. However, some issues should be carefully considered before publication.

*Response: On behalf of all authors, I would like to thank Referee 1 for his/her time, helpful and overall enthusiastic comments, which are very helpful for the improvement of our manuscript. All comments have been taken into account and we attempted to answer all questions.*

**Specific comments:**

1. Introduction: Highlight the advantages of biochar and oyster shell on the immobilization of Cd in soils.

*Response: Thank you for your advice, which is very helpful for the improvement of our manuscript. The advantages of biochar and oyster shell will be added in our revised manuscript. Biochar derived from bio-wastes has widely recommended as a soil amendment in Cd contaminated soil. Amounts of nutrients (such as C, N, P, K, and Mg*

*etc.) in biochar could improve soil fertility and promote plant growth. Moreover, biochar has a large surface area and plenty of functional groups, which are reactive to immobilize heavy metals. Oyster shell is a low-cost and largely available bio-waste product from oyster farming, which is a promising slow-release alkaloid has outstanding effects on pH adjustment and Cd immobilization in soils. Moreover, oyster shell as a low-cost product that can be largely applied in farmland.*

2. L70 functional groups, please specify it.

***Response:** The functional groups on the surface of biochar mainly include -OH, -COOH, C-O, C=O and C-H et al. The functional groups were will added in our revised manuscript.*

3. M&M: L107, I think that there should add a sentence "The main properties of biochar and oyster shell were presented in Table S1.".

***Response:** Thank you for your kind advice. The sentence "The main properties of biochar and and oyster shell were presented in Table S1." will be added in our revised manuscript.*

4. M&M: More details should be provided in the experimental setup, such as water management.

***Response:** Thank you for your kind advice. Detail experimental method will be provided in our revised manuscript.*

5. M&M: L133, Full name of the AAS abbreviation should be provided in the first time.

***Response:** The full name of the AAS abbreviation (Atomic Absorption Spectroscopy) will be provided in our revised manuscript.*

6. M&M: L141-142, The format of references was incorrect, please check.

***Response:** We will carefully check the format of references, which will meet the requirements of this journal.*

7. L175: I think that this part should be "Results" rather than "Results and discussion" because "Results" and "Discussion" in this manuscript were separated.

***Response:** We are sorry for our carelessness. This mistake will be corrected in our revised manuscript.*

8. Results 3.1: The Cd concentration in farmland soils should be addressed.

*Response: The Cd concentration in farmland soils is 0.8 - 1.2 mg/kg. The Cd concentration in farmland soils will be addressed in our revised manuscript.*

9. Result 3.5: Why authors determine the dehydrogenase, urease, acid phosphate and β-galactosidase but not other soil enzyme?

*Response: Dehydrogenase, urease, acid phosphate and β-galactosidase were the important indexes to reflect the soil biological properties. Dehydrogenase unusually reflects the microbial degradation capacity for organic matter. Urease was often used as a biochemical indicator to reflect soil fertility, which played a crucial role on soil nitrogen mineralization. Acid phosphatase plays a vital factor in controlling P mineralization and it reflects the capacity of P mineralization potential in soil. β-galactosidase can catalyze the cellulose into glucose, which play an important role in the microbial glycometabolism. In our revised manuscript, the reasons why choose these enzymes will be carefully explained.*

10. Discussion: The discussion of immobilization mechanisms of Cd can be addressed according to literatures.

*Response: Thank you for your kind advice. The immobilization mechanisms of Cd will be carefully addressed according to literature in our revised manuscript.*

11. L325-326: Authors should provide the composition of oyster shell.

*Response: Thank you for your kind advice. The composition of oyster shell will be added in our revised manuscript.*

12. Figure 6a: The SD might be incorrect, please check.

*Response: Thank you for your kind advice. The incorrect SD will be revised in our revised manuscript.*

13. References: The format of some literature was incorrect. Such as: Line 412; Line 416

*Response: Thank you for your kind advice. The format of literature will be revised according to the requirement of Soil.*

---

## Author Comment (AC2)

**Soil**

**Manuscript No.:** SOIL-2021-145

**Manuscript title:** The cooperative application of oyster shell and biochar efficiently enhanced in-situ remediation of cadmium contaminated soil around intensive industry

**Article type:** Research paper

**Authors:** Bin Wu, Jia Li, Mingping Sheng, He Peng, Dinghua Peng, Heng Xu

**Reviewer 2:**

General comment:

The manuscript discusses the effects of biochar and oyster shell, added into rice-oilseed rape system, on the remediation of cadmium in contaminated soil and reduction of the element in the crops. This manuscript also evaluated the potential health risk from cadmium in the crops grown in the amended soils. This is quite a good piece of work; however, this paper largely lacks coherence with the objective(s), and needs more attention to its writing style and discussion in support of the results observed. This manuscript mainly emphasizes that the cooperative application of biochar and oyster shell is the most effective means for remediating cadmium in contaminated soil, in terms of reducing cadmium bioavailability in soil and cadmium uptake by plants as well as improving a number of soil quality parameters; however, these are indeed not reflected by the results (data/figures). Rather, the sole application of oyster shell, the cheaper one, appeared to be more effective compared to the joint application of biochar and oyster shell. Major revisions are necessary in many cases (please see the specific comments). The discussion section must be re-written, focusing on well constructive explanations; this part should be coherent and readable. In addition, all the references need to be corrected following the journal style. This manuscript may not be considered for publication; however, for a positive decision, significant improvements, and justification with the results in the highlighted contexts must be done, carefully considering the specific comments through major revisions.

> *Response: On behalf of all authors, I would like to thank Referee 2 for his/her time, helpful and overall enthusiastic comments, which are very helpful for the improvement*

*of our manuscript. All comments have been taken into account and we attempted to answer all questions. In this study, we investigated that the effects of biochar and oyster shell on Cd uptake by crops and soil biochemical properties. This is an interesting and practical study, which revealed the cooperative application of biochar and oyster shell both reduced the Cd bioavailability, Cd uptake by crops and improved soil biochemical properties including pH, organic matter, available nutrients and enzyme activities. Maybe the cooperative effects were not clearly expressed in our manuscript. According to research results, oyster shell showed a better effect on reducing Cd bioavailability compared to biochar. For Cd immobilization, there was no significant difference between PA2 (15000 kg/ha oyster shell) and PA3 (7500 kg/ha biochar and 7500 kg/ha oyster shell). However, compared to signal biochar treatment, the cooperative application of oyster shell and biochar could significantly increase the Cd immobilization and decrease the remediation cost. Meanwhile, compared to signal oyster shell treatment, the cooperative application of oyster shell and biochar could significantly enhance the soil biochemical properties, such as the organic matter, Olsen-K, and the activities of dehydrogenase, acid phosphate and β-galactosidase. Therefore, the cooperative application of biochar and oyster shell might be a practical way to immobilize Cd and enhance the soil biochemical properties. In our revised manuscript, we will carefully revise the deficiency according to the Referee's comments. The results and discussion will also be carefully revised. In addition, all comments are easy to be revised in our revised manuscript.*

**Specific comments**

**Abstract:**

Line 30-32. Please rephrase the sentence.

**Response:** *Thank you for your advice. This sentence will be revised in our revised manuscript.*

Lines 32-38. Please rephrase this part, with clear treatment combination. It seems that the treatments were applied twice for the two different crops, but this was not the case.

The repetition of "followed with" should be avoided.

*Response: Thank you for your kind advice. The treatment combination will be clearly rephrased in our revised manuscript.*

Line 38. Please indicate what the HOAc-extractable Cd means (like, bioavailable or so). Please use a single style for stating "HOAc-extractable Cd". This is written in a different way at the line numbers 137 (acetic acid), 197, 315….. Also, what is the significance of only mentioning about Cd reduction in R-PA2 in the abstract?

*Response: Thank you for your kind advice. We will use a signal style for stating "HOAc-extractable Cd" in our revised manuscript.*

Line 38. … compared to control.

*Response: Thank you for your kind advice. This sentence will be revised in our revised manuscript.*

Line 38-44. What is the significance of these observations – please link.

*Response: Thank you for your kind advice. The significance between different treatments will be clearly written in our revised manuscript.*

Line 45. Joint, not jiont. Please check this spelling in other places also.

*Response: We are sorry for our carelessness. This incorrect word will be revised in our revised manuscript.*

Line 47. How is the joint application more effective than the sole applications of biochar and oyster shell, while the results don't show any significant differences?

*Response: We are sorry for our unclear clarification. According to research results, oyster shell showed a better effect on reducing Cd bioavailability compared to biochar. For Cd immobilization, there was no significant difference between PA2 (15000 kg/ha oyster shell) and PA3 (7500 kg/ha biochar and 7500 kg/ha oyster shell). However, compared to signal biochar treatment, the cooperative application of oyster shell and biochar could significantly increase the Cd immobilization and decrease the*

*remediation cost. Meanwhile, compared to signal oyster shell treatment, the cooperative application of oyster shell and biochar could significantly enhance the soil biochemical properties, such as the organic matter, Olsen-K, and the activities of dehydrogenase, acid phosphate and β-galactosidase. Therefore, the cooperative application of biochar and oyster shell might be a practical way to immobilize Cd and enhance the soil biochemical properties.*

**Introduction**

Line 52-53. Please rephrase the sentence, appropriately linking the second part.

*Response: Thank you for your kind advice. This sentence will be rephrased in our revised manuscript.*

Line 56. … has been ranked ….

*Response: Thank you for your kind advice. This sentence will be rephrased in our revised manuscript.*

Line 66. In recent years, …..bio-wastes has been widely ….

*Response: Thank you for your kind advice. This sentence will be rephrased in our revised manuscript.*

Line 68. Please use either 'and' or 'etc'.

*Response: Thank you for your kind advice. This sentence will be revised in our revised manuscript.*

Line 71. Limited or limits?

*Response: Thank you for your kind advice. This word will be revised in our revised manuscript.*

Line 72-74. Please rephrase this sentence. What is meant by "…not very significant"? Please write acidic instead of acid.

*Response:* *Thank you for your kind advice. This sentence will be rephrased in our revised manuscript.*

Line 76-78. Please rephrase this sentence. 'that' to be removed, and 'which' to be added between alkaloid and has.

*Response:* *Thank you for your kind advice. This sentence will be rephrased in our revised manuscript.*

Line 81. There is little known….

*Response:* *Thank you for your kind advice. This sentence will be rephrased to better understanding in our revised manuscript.*

Line 83. … are the main food…

*Response:* *Thank you for your kind advice. This sentence will be corrected in our revised manuscript.*

Line 85-87. Please give references for the previous studies. Please write "rice-oilseed rape".

*Response:* *Thank you for your kind advice. We will introduce more previous studies to highlight our research significance. The "rice-oilseed rape" will be corrected in our revised manuscript.*

Line 87. Based on the above…

*Response:* *Thank you for your kind advice. This sentence will be rephrased in our revised manuscript.*

Line 87. … a field experiment, not filed.

*Response:* *We are sorry for our carelessness. This word will be corrected in our revised manuscript.*

Line 92. Alkeline-N or Alkaline-N? Please check this throughout the manuscript.

*Response:* *We are sorry for our carelessness. This word will be corrected in our revised*

*manuscript.*

Line 93. PCA is generally used for Principal Component Analysis. PCA here should be avoided.

*Response: Thank you for your kind advice. We will re-analyze the correlation of different factors in our revised manuscript.*

**Materials and methods**

Line 103. Table S1 is missing.

*Response: We are sorry for our carelessness. This word will be corrected in our revised manuscript.*

Line 115 & 119. "passivators" may be replaced with a more appropriate word. However, the 'passivators' are not specified before.

*Response: Thank you for your kind advice. This word will be rephrased in our revised manuscript.*

Line 137. Please see the comment made before; compare with the line no. 38….

*Response: Thank you for your kind advice. This sentence will be rephrased in our revised manuscript.*

Line 141. Alkeline-N or Alkaline-N? Please check the reference Liu et al. 2017 to confirm its content about the determination of alkaline-N.

*Response: Thank you for your kind advice. We will carefully check this reference.*

Line 142. TOC and OM should be indicated in full within (). (Walz et al., 2017) should be written as Walz et al. (2017). Please follow this style for other instances; e.g., in line no. 141.

*Response: Thank you for your kind advice. We will carefully check and revise the format of all references in our manuscript.*

Line 155. Human health risk….

*Response:* *Thank you for your kind advice. This sentence will be rephrased in our revised manuscript.*

Line 155. Please use a single format – either 'Cd' or 'cadmium' throughout the manuscript.

*Response:* *Thank you for your kind advice. We will carefully check the format throughout the manuscript.*

Line 156. The health risks of Cd to adults and children in the crops… - this part of the sentence should be revised.

*Response:* *Thank you for your kind advice. This sentence will be rephrased in our revised manuscript.*

Line 157. Please define Hazard Quotient – what it indicates actually.

*Response:* *Thank you for your kind advice. We will clearly explain the Hazard Quotient.*

Line 158. Please write HQ in full when it is in the beginning of a sentence. Please follow this for other cases throughout the manuscript.

*Response:* *Thank you for your kind advice. The full name of HQ will be explained at the beginning of the sentence in our revised manuscript.*

Line 159. calculated using the following equation:

*Response:* *Thank you for your kind advice. This sentence will be rephrased in our revised manuscript.*

Line 160. (1) can be removed.

*Response:* *Thank you for your kind advice. This sentence will be rephrased in our revised manuscript.*

Line 161. Please write "where," beforehand.

*Response:* *Thank you for your kind advice. This sentence will be rephrased in our revised manuscript.*

Line 174. What is actually meant by "all statistics"? Should it be graphical interpretation or so…?

*Response: Thank you for your kind advice. Data statistical significance was analyzed using SPSS 18.0 package, and means values were considered to be different when P < 0.05 using least significant difference (LSD). Figures were performed using Origin 8.0 (USA).*

Results

Line 175. 'Results' only.

*Response: We are sorry for our carelessness. This mistake will be correct in our revised manuscript.*

Line 180. OM is generally expressed in %.

*Response: Thank you for your kind advice. This expression of OM will be corrected in our revised manuscript.*

Line 182. Please rephrase. What is indicated by 'reached' here?

*Response: Thank you for your kind advice. The sentence will be rephrased in our revised manuscript.*

Line 197. Please see the comment made before; compare with the line no. 38….

*Response: Thank you for your kind advice. The sentence will be rephrased in our revised manuscript.*

Line 197-198. The second part of the sentence should be in the discussion section. Please use reference(s) for such statement.

*Response: This result will be adequately discussed according to references.*

Line 200. … flooding irrigation… Please use one.

*Response: Thank you for your kind advice. The sentence will be rephrased in our*

*revised manuscript.*

Line 212-213. Not true. Please rephrase with actual observations from the figure.

*Response: we are sorry for our carelessness. We will carefully check and revise this sentence according to data.*

Line 213-214. Please rephrase "treatments without amendments". Please check the Figure 3a to confirm the value 0.88 mg/kg.

*Response: We are sorry for our carelessness. We will carefully check and revise this sentence according to data.*

Line 216. … oyster shell had the ….

**Response:** *Thank you for your kind advice. The sentence will be rephrased in our revised manuscript.*

Line 220. Oilseed Cd content in RT-PA3 was not significantly lower than that in the Control. Please check and amend.

*Response: Thank you for your kind advice. We will carefully check and revise this sentence.*

Line 221. "The results indicated that biochar and oyster shell application…" – sole application or combine application?

*Response: We are sorry for our unclear statement. This sentence should be "The results indicated that the cooperative application of biochar and oyster shell could efficiently decrease Cd accumulation in brown rice and oilseed".*

Line 224. Please write HQ in full as in the beginning of the sentence.

*Response: Thank you for your kind advice. The full name of HQ will be explained at the beginning of the sentence in our revised manuscript.*

Line 224-225. This sentence can be removed. Rather, 'Figure 4' can be written at the end of the sentence in line no. 227.

*Response: Thank you for your kind advice. This sentence will be removed in the revised manuscript.*

Line 226. …. adult female > adult male, which indicated that children were….

*Response: Thank you for your kind advice. The sentence will be rephrased in our revised manuscript.*

Line 226. Please use a more appropriate word for 'sensitive'.

*Response: Thank you for your kind advice. The word will be replaced using a more* appropriate word *in our revised manuscript.*

Line 228. … HQ values of consuming brown rice for adult male, adult female and children reached 5.46, 6.21 and 6.82, respectively. – What does it mean by "reached"? Is this the max value for a certain group? Moreover, the values given here do not match with the observations in the Figure 4a. Please check and amend clearly.

*Response: We are sorry for our unclear writing. The values were the average mean. This sentence will be revised to better understand in our revised manuscript.*

Line 229. … HQ values for brown rice intake…

*Response: Thank you for your kind advice. The sentence will be rephrased in our revised manuscript.*

Line 229-231. Please rephrase. The decrease in the HQ values for which group(s) – adult male female or children?

*Response: Thank you for your kind advice. The sentence will be rephrased in our revised manuscript.*

Line 231-233. Please rephrase. What does the "significantly" mean here?

*Response: Thank you for your kind advice. The sentence will be rephrased to better understanding in our revised manuscript.*

Line 237. It should be Figure 5a, not 2a.

*Response:* *We are sorry for our mistake. This mistake will be corrected in our revised manuscript.*

Line 239. Please avoid using 'points' for indicating the pH values.

*Response:* *Thank you for your kind advice. This word will be replaced in our revised manuscript.*

Line 240. slightly, not slight.

*Response:* *We are sorry for our carelessness. This mistake will be corrected in our revised manuscript.*

Line 241. CEC of soil….

*Response:* *Thank you for your kind advice. The sentence will be rephrased in our revised manuscript.*

Line 242. Figure 5b, not 2b.

*Response:* *We are sorry for our carelessness. This mistake will be corrected in our revised manuscript.*

Line 244-245. This is irrelevant here.

*Response:* *We are sorry for our carelessness. This mistake will be corrected in our revised manuscript.*

Line 247. Table S2 is missing.

*Response:* *We are sorry for our carelessness. This mistake will be corrected in our revised manuscript.*

Line 260-262. Please rephrase.

*Response:* *Thank you for your kind advice. The sentence will be rephrased in our revised manuscript.*

Line 262-264. Please rephrase.

*Response: Thank you for your kind advice. The sentence will be rephrased in our revised manuscript.*

Line 265-266. Please rephrase.

*Response: Thank you for your kind advice. The sentence will be rephrased in our revised manuscript.*

Line 267. …. biochar had no negative …

*Response: Thank you for your kind advice. The sentence will be rephrased in our revised manuscript.*

Line 269-270. Please make it clear whether it was individual application of the treatments or combined application.

*Response: Thank you for your kind advice. The sentence will be rephrased to better understanding in our revised manuscript.*

Line 274. Please rephrase "to experimental data".

*Response: Thank you for your kind advice. The sentence will be rephrased in our revised manuscript.*

Line 275. Person correlation value is expressed by 'r'. Please check this throughout the manuscript.

*Response: Thank you for your kind advice. We will carefully check the statement.*

Line 278. The Figure 7b….

*Response: We are sorry for our carelessness. This mistake will be corrected in our revised manuscript.*

Line 278. …. Cd uptake by oilseed rape ….

*Response: Thank you for your kind advice. The sentence will be rephrased in our revised manuscript.*

Line 282. Acidic

*Response: Thank you for your kind advice. This mistake will be corrected in our revised manuscript.*

Line 287. (detailed information see Supplementary Information) – should be written in an appropriate way. There is no SI attached with the manuscript.

*Response: Thank you for your kind advice. We will revise this sentence in an appropriate way.*

Line 288. The dosage of application was 15 t/ha, i.e., 15000 kg/ha. So, please review this amount and also the calculated cost in line numbers 289 & 290.

*Response: Thank you for your kind advice. We will re-calculate the cost by the price and amount.*

Line 289-290. joint

*Response: Thank you for your kind advice. This mistake will be corrected in our revised manuscript.*

Line 291. Please amend the sentence as '… biochar and oyster shell was found to be more effective….'. Again, how this was more effective than the individual applications? The use of oyster shell alone had either similar effects or its effects were not significantly different than the joint application. Moreover, oyster shell is much cheaper than biochar, and thus, compared to the combined use.

*Response: Thank you for your kind advice. The sentence will be rephrased in our revised manuscript. Other indexes should be also considered. In our revised manuscript, we will explain this sentence more clearly.*

Line 292-294. Should be in the discussion section.

*Response: Thank you for your kind advice. The sentence will be replaced to the discussion section.*

**Discussion**

Line 296. …. are the most …

*Response:* *Thank you for your kind advice. The sentence will be rephrased in our revised manuscript.*

Line 303-304. Please use a reference for this statement.

*Response:* *Thank you for your kind advice. We will add a reference for this statement.*

Line 305. …. have…

*Response:* *Thank you for your kind advice. This mistake will be corrected in our revised manuscript.*

Line 306. … had …

*Response:* *Thank you for your kind advice. This mistake will be corrected in our revised manuscript.*

Line 306. Please give references for the previous studies.

*Response:* *Thank you for your kind advice. We will add references for this statement.*

Line 307. Please rephrase the sentence.

*Response:* *Thank you for your kind advice. The sentence will be rephrased in our revised manuscript.*

Line 310. But the reductions in Cd were not statistically significant compared to the control treatment.

*Response:* *Thank you for your kind advice. The sentence will be rephrased in our revised manuscript.*

Line 311-314. Please rephrase.

*Response:* *Thank you for your kind advice. The sentence will be rephrased in our revised manuscript.*

Line 315. Please use an appropriate word for "has" here.

*Response:* *Thank you for your kind advice. We will use an appropriate word here.*

Line 315-326. Please rephrase with constructive discussion in the context.

*Response: Thank you for your kind advice. This part will be carefully rephrased in our revised manuscript.*

The discussions are mostly linked to the pH; however, the pH for oyster shell and combined treatment are not significantly different, and so the effects due to pH as well. How can this strongly support the joint application over oyster shell alone? The authors can also highlight the benefits of using biochar with oyster shell linking with different other factors.

*Response: Thank you for your kind advice. According to research results, oyster shell showed a better effect on reducing Cd bioavailability compared to biochar. For Cd immobilization, there was no significant difference between PA2 (15000 kg/ha oyster shell) and PA3 (7500 kg/ha biochar and 7500 kg/ha oyster shell). However, compared to signal biochar treatment, the cooperative application of oyster shell and biochar could significantly increase the Cd immobilization and decrease the remediation cost. Meanwhile, compared to signal oyster shell treatment, the cooperative application of oyster shell and biochar could significantly enhance the soil biochemical properties, such as the organic matter, Olsen-K, and the activities of dehydrogenase, acid phosphate and β-galactosidase. Therefore, the cooperative application of biochar and oyster shell might be a practical way to immobilize Cd and enhance the soil biochemical properties. In our revised manuscript, we will clearly state the benefits of using biochar with oyster shell with different factors.*

Line 337. Please write 'P' in full in the beginning of the sentence.

*Response: Thank you for your kind advice. The full name will be added at the beginning of the sentence.*

Line 345. Please avoid the use of "PCA".

*Response: Thank you for your kind advice. We will use another method to analyze the correlation of the different factors.*

Line 350-352. Please rephrase.

*Response: Thank you for your kind advice. This sentence will be carefully rephrased in our revised manuscript.*

Line 354. Please use appropriate words for 'obvious' and 'stimulation'.

*Response: Thank you for your kind advice. We will use appropriate words to replace to these words.*

Line 354-355. Please either remove this sentence or rephrase.

*Response: Thank you for your kind advice. This sentence will be carefully rephrased in our revised manuscript.*

Line 357. Please add year for the reference used, and remove the reference written at the end of the sentence in the line no. 358.

*Response: Thank you for your kind advice. We will revise the format of references correctly.*

Conclusions

Lines 372. Please use a more appropriate for 'extraordinary' here?

*Response: Thank you for your kind advice. We will use appropriate word to replace this word.*

Line 378. …. our, not Our.

*Response: We are sorry for our carelessness. This mistake will be corrected in our revised manuscript.*

**Others**

Figure 7. The values on the figures are not clear.

*Response: Thank you for your kind advice. We will revise the figures to be clear.*

**References**

Please follow the journal style of referencing in the reference section. All the references need to be corrected following the journal style. So, please check and amend throughout the reference section.

**Response:** *Thank you for your kind advice. We will carefully revise the format of references according to the style of journal.*

---

## Author Response (AR1)

**Response to the reviewers' comments**

**Soil**

**Manuscript No.:** SOIL-2021-145

**Manuscript title:** The cooperative application of oyster shell and biochar efficiently enhanced in-situ remediation of cadmium contaminated soil around intensive industry

**Article type:** Research paper

**Authors:** Bin Wu, Jia Li, Mingping Sheng, He Peng, Dinghua Peng, Heng Xu

**Reviewer 1:**

The authors of the manuscript Soil-2021-145 "The cooperative application of oyster shell and biochar efficiently enhanced in-situ remediation of cadmium contaminated soil around intensive industry" addressed an interesting and important topic by using amendments (biochar and oyster shell) to achieve the safe-production of crops in the Cd contaminated soil following the rice-oilseed rape rotation. Indicators reflected soil quality including soil fertility and enzyme activities were also investigated by authors. Cd contaminant in farmland has been causing a great concern on human health. Overall, the authors present a well-designed study with appropriate methods, and this study provided a practical method to reduce the Cd contents in crops and reduce the remediation cost. I think that this study is well-worth to be published. However, some issues should be carefully considered before publication.

*Response: On behalf of all authors, I would like to thank Referee 1 for his/her time, helpful and overall enthusiastic comments, which are very helpful for the improvement of our manuscript. All comments have been taken into account and we attempted to answer all questions.*

**Specific comments:**

➢ 1. Introduction: Highlight the advantages of biochar and oyster shell on the immobilization of Cd in soils.

*Response: Thank you for your kind advice, which is very helpful for the improvement of our manuscript. The advantages of biochar and oyster shell have been added in our revised manuscript. Biochar derived from bio-wastes has been*

*widely recommended as a promising soil amendment for the Cd contaminated soil. Amounts of nutrients (such as C, N, P, K, and Mg etc.) in biochar could improve soil fertility and promote plant growth. Moreover, biochar has a large surface area and plenty of functional groups, which are reactive to immobilize heavy metals. Oyster shell is a low-cost and largely available bio-waste product from oyster farming, which is a promising slow-release alkaloid has outstanding effects on pH adjustment and Cd immobilization in acidic soils. Moreover, oyster shell as a low-cost product that can be largely applied in farmland.* **(Introduction: Line 66-77)**

➢ 2. L70 functional groups, please specify it.

   **Response:** *The functional groups on the surface of biochar mainly include -OH, -COOH, C-O, C=O and C-H. The functional groups have been added in our revised manuscript.* **(Introduction: Line 70)**

➢ 3. M&M: L107, I think that there should add a sentence "The main properties of biochar and oyster shell were presented in Table S1.".

**Response:** *Thank you for your kind advice. The sentence "The main properties of biochar and oyster shell were presented in Table S1." has been added in our revised manuscript.* **(Introduction: Line 106-107)**

➢ 4. M&M: More details should be provided in the experimental setup, such as water management.

   **Response:** *Thank you for your kind advice. Detail experimental method has been provided in our revised manuscript.* **(Material and methods: Line 115-122)**

➢ 5. M&M: L133, Full name of the AAS abbreviation should be provided in the first time.

   **Response:** *Thank you for your kind advice. The full name of the AAS abbreviation "Atomic Absorption Spectroscopy" has been provided in our revised manuscript.* **(Material and methods: Line 128)**

➢ 6. M&M: L141-142, The format of references was incorrect, please check.

   **Response:** *Thank you for your kind advice. We have carefully checked the format of references, which meet the requirements of this journal.* **(Whole manuscript)**

➤ 7. L175: I think that this part should be "Results" rather than "Results and discussion" because "Results" and "Discussion" in this manuscript were separated.

*Response: We are sorry for our carelessness. This mistake has been corrected in our revised manuscript. **(Results: Line 171)***

➤ 8. Results 3.1: The Cd concentration in farmland soils should be addressed.

*Response: The Cd concentration in farmland soils is 1.6 mg/kg. The Cd concentration in farmland soils has been added in our revised manuscript (Table S1).*

**(Supplementary materials: Table S1)**

➤ 9. Result 3.5: Why authors determine the dehydrogenase, urease, acid phosphate and β-galactosidase but not other soil enzyme?

*Response: Dehydrogenase, urease, acid phosphate and β-galactosidase were the important indexes to reflect the soil biological properties. Dehydrogenase usually reflects the microbial degradation capacity for organic matter. Urease is often used as a biochemical indicator to reflect soil fertility, which plays a crucial role on soil nitrogen mineralization. Acid phosphatase plays a vital factor in controlling P mineralization and it reflects the capacity of P mineralization potential in soil. β-galactosidase can catalyze the cellulose into glucose, which play an important role in the microbial glycometabolism. In our revised manuscript, the reasons why choose these enzymes have been carefully explained in the discussion section. (**Discussion: Line 323-334)***

➤ 10. Discussion: The discussion of immobilization mechanisms of Cd can be addressed according to literatures.

*Response: Thank you for your kind advice. The immobilization mechanisms of Cd have been carefully addressed according to literature in our revised manuscript. **(Discussion: Line 286-299)***

➤ 11. L325-326: Authors should provide the composition of oyster shell.

*Response: Thank you for your kind advice. The composition of oyster shell has been added in our revised manuscript.* **(Supplementary materials: Table S2)**

Table S2 The main composition of oyster shell.

| Main compounds | Al$_2$O$_3$ (%) | CaCO$_3$ (%) | MgO (%) | Na$_2$O (%) | P$_2$O$_5$ (%) | SiO$_2$ (%) |
|---|---|---|---|---|---|---|
| Values | 0.40 | 95.5 | 0.62 | 1.02 | 0.4 | 0.82 |

➢ 12. Figure 6a: The SD might be incorrect, please check.

*Response:* *Thank you for your kind advice. The incorrect SD has been revised in our revised manuscript.* **(Figure 6a)**

➢ 13. References: The format of some literature was incorrect. Such as: Line 412; Line 416

*Response:* *Thank you for your kind advice. The format of literature will be revised according to the requirement of Soil.* **(References)**

**Reviewer 2:**

General comment:

The manuscript discusses the effects of biochar and oyster shell, added into rice-oilseed rape system, on the remediation of cadmium in contaminated soil and reduction of the element in the crops. This manuscript also evaluated the potential health risk from cadmium in the crops grown in the amended soils. This is quite a good piece of work; however, this paper largely lacks coherence with the objective(s), and needs more attention to its writing style and discussion in support of the results observed. This manuscript mainly emphasizes that the cooperative application of biochar and oyster shell is the most effective means for remediating cadmium in contaminated soil, in terms of reducing cadmium bioavailability in soil and cadmium uptake by plants as well as improving a number of soil quality parameters; however, these are indeed not reflected by the results (data/figures). Rather, the sole application of oyster shell, the cheaper one, appeared to be more effective compared to the joint application of biochar and oyster shell. Major revisions are necessary in many cases (please see the specific comments). The discussion section must be re-written, focusing on well constructive explanations; this part should be coherent and readable.

In addition, all the references need to be corrected following the journal style. This manuscript may not be considered for publication; however, for a positive decision, significant improvements, and justification with the results in the highlighted contexts must be done, carefully considering the specific comments through major revisions.

*Response: On behalf of all authors, I would like to thank you for your time, helpful and overall enthusiastic comments, which are very helpful for the improvement of our manuscript. All comments have been taken into account and we attempted to answer all questions. In this study, we investigated the effects of biochar and oyster shell on Cd uptake by crops and soil biochemical properties. This is an interesting and practical study, which revealed the cooperative application of biochar and oyster shell both reduced the Cd bioavailability, Cd uptake by crops and improved soil biochemical properties including pH, organic matter, available nutrients and enzyme activities. However, maybe the cooperative effects were not clearly expressed in our original manuscript. According to the research results, oyster shell showed a better effect on reducing Cd bioavailability than biochar. For Cd immobilization, there was no significant difference between PA2 (15000 kg/ha oyster shell) and PA3 (7500 kg/ha biochar and 7500 kg/ha oyster shell). However, compared to signal biochar treatment, the cooperative application of oyster shell and biochar could significantly increase the Cd immobilization and decrease the remediation cost. Meanwhile, compared to the signal oyster shell treatment, the cooperative application of oyster shell and biochar could markedly enhance the soil biochemical properties, such as the organic matter, available K, and the activities of dehydrogenase, acid phosphate and β-galactosidase. Therefore, the cooperative application of biochar and oyster shell might be a practical way to immobilize Cd and enhance the soil biochemical properties. In our revised manuscript, we have carefully revised the deficiency according to your comments. The results and discussion will also be carefully revised. We think that the quality of our manuscript has been highly improved, which meets the requirements of publication in soil. In our revised manuscript, all changes were marked in red color.*

**Specific comments**

**Abstract:**

➤ Line 30-32. Please rephrase the sentence.

*Response: Thank you for your advice. This sentence has been revised in our revised manuscript as follows: "In this study, we investigated the possibility of the application of biochar and oyster shell to reduce the Cd uptake by crops and modify the soil biochemical properties.".* (*Abstract: Line 29-30*)

➤ Lines 32-38. Please rephrase this part, with clear treatment combination. It seems that the treatments were applied twice for the two different crops, but this was not the case. The repetition of "followed with" should be avoided.

*Response: Thank you for your kind advice, which is very helpful for the improvement of our manuscript. The treatment combination has been clearly rephrased in our revised manuscript as follows: "A filed study based on the rice-oilseed rape rotation was done and the treatments were comprised of without amendments (PA0), 15000 kg/ha biochar (PA1), 15000 kg/ha oyster shell (PA2), and 7500 kg/ha biochar and 7500 kg/ha oyster shell (PA3). ".* (*Abstract: Line 30-33*)

➤ Line 38. Please indicate what the HOAc-extractable Cd means (like, bioavailable or so). Please use a single style for stating "HOAc-extractable Cd". This is written in a different way at the line numbers 137 (acetic acid), 197, 315….. Also, what is the significance of only mentioning about Cd reduction in R-PA2 in the abstract?

*Response: Thank you for your kind advice. In our revised manuscript, we used a signal style for stating "HOAc-extractable Cd". In addition, the reduction of HOAc-extractable Cd in R-PA2 was mentioned because the maximum reduction of HOAc-extractable Cd was observed in R-PA2. In our revised manuscript, this sentence was revised to better understanding as follows: "Compared to PA0, the HOAc-extractable Cd in the PA1, PA2 and PA3 treatments was reduced by 4.76 - 20.79%, 17.86 - 38.61% and 5.95 - 10.89%, respectively. ".* (*Abstract: Line 35-36*)

➢ Line 38. … compared to control.

*Response: Thank you for your kind advice. In our revised manuscript, the sentence has been revised to better understanding as follows: "The cooperative application of biochar and oyster shell reduced the Cd accumulation in brown rice and oilseed by 29.67% and 19.74%, respectively, compared to control.".* (*Abstract: Line 37-38*)

➢ Line 38-44. What is the significance of these observations – please link.

*Response: Thank you for your kind advice. The significance between different treatments has been clearly re-written in our revised manuscript. Means values were considered to be significant different when $P < 0.05$ using least significant difference (LSD). The revised manuscript is: "In addition, the available P in the PA2 and PA3 treatments was significantly ($p < 0.05$) increased by 200.96 - 295.92% and 187.46 - 280.04% compared with that in the PA0 treatments.".* (*Abstract: Line 40-42*)

➢ Line 45. Joint, not jiont. Please check this spelling in other places also.

*Response: We are sorry for our carelessness. This mistake has been revised in our revised manuscript.* (*Abstract: Line 43*)

➢ Line 47. How is the joint application more effective than the sole applications of biochar and oyster shell, while the results don't show any significant differences?

*Response: We are sorry for our unclear clarification. According to research results, oyster shell showed a better effect on reducing Cd bioavailability than biochar. For Cd immobilization, there was no significant difference between PA2 (15000 kg/ha oyster shell) and PA3 (7500 kg/ha biochar and 7500 kg/ha oyster shell). However, compared to signal biochar treatment, the cooperative application of oyster shell and biochar could significantly increase the Cd immobilization and decrease the remediation cost. Meanwhile, compared to signal oyster shell treatment, the cooperative application of oyster shell and biochar could significantly enhance the soil biochemical properties, such as the organic matter, available K, and the activities*

of dehydrogenase, acid phosphate and β-galactosidase. Therefore, the cooperative application of biochar and oyster shell might be a practical way to immobilize Cd and enhance the soil biochemical properties. In our revised manuscript, we have carefully clarified the significance of the cooperative application of biochar and oyster shell. For better understanding, this sentence has been revised to: "These results demonstrated that the utilization of biochar and oyster shell might be an effective pathway to reduce the Cd uptake by crops and improve soil biochemical properties.".

(**Abstract: Line 46-48**)

**Introduction**

➢ Line 52-53. Please rephrase the sentence, appropriately linking the second part.

*Response: Thank you for your kind advice. The sentence will be rephrased in our revised manuscript as follows: "Cadmium (Cd) contamination of agricultural soils is a worldwide environmental problem, which has been seriously threatening to human health (Yang et al., 2021). The excessive intake of Cd by food chain can cause serious damages to bones, thyroid, and kidneys (Ma et al., 2021)".* (**Introduction: Line 52-55**)

*References:*

*Ma, Y., Ran, D., Shi, X., Zhao, H., Liu, Z., 2021. Cadmium toxicity: A role in bone cell function and teeth development. Science of The Total Environment 769, 144646.*

*Yang, Y., Li, Y., Dai, Y., Wang, M., Chen, W., Wang, T., 2021. Historical and future trends of cadmium in rice soils deduced from long-term regional investigation and probabilistic modeling. Journal of Hazardous Materials 415, 125746.*

➢ Line 56. … has been ranked ….

*Response: We are sorry for this mistake. This sentence has been rephrased in our revised manuscript as follows: "According to the latest national survey on the status of soil environmental quality in China, Cd has been ranked as the highest contaminants (7%) among all heavy metals (Mou et al., 2020)".* **(Introduction: Line**

*55-57*)

*References:*

*Mou, H., Chen, W., Xue, Z., Li, Y., Ao, T., Sun, H., 2020. Effect of irrigation water system's distribution on rice cadmium accumulation in large mild cadmium contaminated paddy field areas of Southwest China. Science of The Total Environment 746, 141248.*

➢ Line 66. In recent years, …..bio-wastes has been widely ….

*Response: Thank you for your kind advice. This sentence has been rephrased in our revised manuscript as follows: "Biochar derived from bio-wastes has been widely recommended as a great soil amendment (Zong et al., 2021)".* **(Introduction: Line 66-67**)

*References:*

*Zong, Y., Xiao, Q., Lu, S., 2021. Biochar derived from cadmium-contaminated rice straw at various pyrolysis temperatures: Cadmium immobilization mechanisms and environmental implication. Bioresource Technology 321, 124459.*

➢ Line 68. Please use either 'and' or 'etc'.

*Response: Thank you for your kind advice. This mistake has been corrected in our revised manuscript. The new sentence is: "Amounts of nutrients (such as C, N, P, K, and Mg) in biochar can improve soil fertility and promote plant growth (Lu et al., 2015)."* **(Introduction: Line 68**)

*References:*

*Lu, H., Li, Z., Fu, S., Méndez, A., Gascó, G., Paz-Ferreiro, J., 2015. Combining phytoextraction and biochar addition improves soil biochemical properties in a soil contaminated with Cd. Chemosphere 119, 209-216.*

➢ Line 71. Limited or limits?

*Response: Thank you for your kind advice. This word has been revised in our revised*

*manuscript. The new sentence is: "However, the high price of biochar limits its large application.".* **(Introduction: Line 71-72)**

➢ Line 72-74. Please rephrase this sentence. What is meant by "…not very significant"? Please write acidic instead of acid.

*Response: Thank you for your kind advice. This sentence has been rephrased in our revised manuscript as follows: "The application of biochar can not effectively change soil pH in acidic fields".* **(Introduction: Line 72-73)**

➢ Line 76-78. Please rephrase this sentence. 'that' to be removed, and 'which' to be added between alkaloid and has.

*Response: Thank you for your kind advice. This sentence has been rephrased in our revised manuscript as follows: "Oyster shell is a promising slow-release alkaloid, which has outstanding effects on pH adjustment and Cd immobilization in soils".* **(Introduction: Line 76-77)**

➢ Line 81. There is little known….

*Response: Thank you for your kind advice. This sentence has been removed in our revised manuscript.*

➢ Line 83. … are the main food…

*Response: Thank you for your kind advice. This sentence has been rephrased in our revised manuscript as follows: "Rice and oilseed rape are the main food and economic crops in southwest China, and rice-oilseed rape rotation is the dominant production model (Liu et al., 2014)".* **(Introduction: Line 81)**

*References:*

*Liu, H.-B., Gou, Y., Wang, H.-Y., Li, H.-M., Wu, W., 2014. Temporal changes in climatic variables and their impact on crop yields in southwestern China. International Journal of Biometeorology 58, 1021-1030.*

➤ Line 85-87. Please give references for the previous studies. Please write "rice-oilseed rape".

*Response: Thank you for your kind advice. This sentence has been rephrased in our revised manuscript and the references were added. The "rice-oilseed rape" was corrected in our revised manuscript. The new sentence is: "Previous studies mainly focused on the effects of amendments on reducing the Cd uptake by rice (Tang et al., 2020; Yin et al., 2022), while the remediation efficiency of passivators under the rice-oilseed rape rotation was little known.".* **(Introduction: Line 84)**

*References:*

*Tang, X., Shen, H., Chen, M., Yang, X., Yang, D., Wang, F., Chen, Z., Liu, X., Wang, H., Xu, J., 2020. Achieving the safe use of Cd- and As-contaminated agricultural land with an Fe-based biochar: A field study. Sci Total Environ 706, 135898.*

*Yin, Z., Sheng, H., Xiao, H., Xue, Y., Man, Z., Huang, D., Zhou, Q., 2022. Inter-annual reduction in rice Cd and its eco-environmental controls in 6-year biannual mineral amendment in subtropical double-rice cropping ecosystems. Environmental Pollution 293, 118566.*

➤ Line 87. Based on the above…

*Response: Thank you for your kind advice. This sentence has been rephrased in our revised manuscript as follows: "Based on the above opinions, a field experiment under the rice-oilseed rape rotation was designed.".* **(Introduction: Line 85)**

➤ Line 87. … a field experiment, not filed.

*Response: We are sorry for our carelessness. This mistake has been corrected in our revised manuscript as follows: "Based on the above opinions, a field experiment under the rice-oilseed rape rotation was designed.".* **(Introduction: Line 86)**

➤ Line 92. Alkeline-N or Alkaline-N? Please check this throughout the manuscript.

*Response: We are sorry for our carelessness. This mistake was corrected in our revised manuscript.* **(Whole manuscript)**

➢ Line 93. PCA is generally used for Principal Component Analysis. PCA here should be avoided.

*Response: Thank you for your kind advice. We revised the statement in our revised manuscript. According to previous studies (Guangming et al., 2017; Yong et al., 2022), the correlation between Cd contents in crops, available Cd, soil pH, soil CEC, available N, available P, available K, organic carbon and the activities of soil enzyme by the Pearson Correlation Analysis.* **(Introduction: Line 92)**

*References:*

*Guangming, L., Xuechen, Z., Xiuping, W., Hongbo, S., Jingsong, Y., Xiangping, W., 2017. Soil enzymes as indicators of saline soil fertility under various soil amendments. Agriculture, Ecosystems & Environment 237, 274-279.*

*Yong, Y., Xu, Y., Huang, Q., Sun, Y., Wang, L., Liang, X., Qin, X., Zhao, L., 2022. Remediation effect of mercapto-palygorskite combined with manganese sulfate on cadmium contaminated alkaline soil and cadmium accumulation in pak choi (Brassica chinensis L.). Science of The Total Environment 813, 152636.*

**Materials and methods**

➢ Line 103. Table S1 is missing.

*Response: We are sorry for our carelessness. The Table S1 was placed in Supplementary Materials.* **(Supplementary Materials: Table S1)**

➢ Line 115 & 119. "passivators" may be replaced with a more appropriate word. However, the 'passivators' are not specified before.

*Response: Thank you for your kind advice. The "passivators" was specified to "biochar and oyster shell".* **(Materials and method: Line 116-117)**

➢ Line 137. Please see the comment made before; compare with the line no. 38….

*Response: Thank you for your kind advice. This sentence was corrected in our revised manuscript as follows: "2 g of soil sample was mixed with 40 mL 0.11 M acetic acid*

*(HOAc) and shaken at 25 ℃, 150 rpm for 16 h.".* **(Materials and method: Line 132-133)**

> Line 141. Alkeline-N or Alkaline-N? Please check the reference Liu et al. 2017 to confirm its content about the determination of alkaline-N.

**Response:** *Thank you for your kind advice. We have carefully checked and revised this sentence as follows: "Available phosphorus (P), available potassium (K), and available nitrogen (N) were measured according to the method described by Wu et al. (2018).".* **(Materials and method: Line 135)**

*References:*

*Wu, B., Hou, S., Peng, D., Wang, Y., Wang, C., Xu, F., Xu, H., 2018. Response of soil micro-ecology to different levels of cadmium in alkaline soil. Ecotoxicol Environ Saf 166, 116-122.*

> Line 142. TOC and OM should be indicated in full within (). (Walz et al., 2017) should be written as Walz et al. (2017). Please follow this style for other instances; e.g., in line no. 141.

**Response:** *Thank you for your kind advice. This sentence has been revised as follows: "Soil TOC and OM were determined by the method described by Walz et al. (2017).".* **(Materials and method: Line 137)**

*References:*

*Walz, J., et al., 2017. Regulation of soil organic matter decomposition in permafrost-affected Siberian tundra soils - Impact of oxygen availability freezing and thawing, temperature, and labile organic matter. Soil Biology & Biochemistry. 110, 34-43.*

> Line 155. Human health risk….

**Response:** *We are sorry for our carelessness. This mistake was corrected in our revised manuscript.* **(Materials and method: Line 150)**

➢ Line 155. Please use a single format – either 'Cd' or 'cadmium' throughout the manuscript.

*Response: Thank you for your kind advice. We have carefully checked the format of Cd throughout the manuscript.* **(Whole manuscript)**

➢ Line 156. The health risks of Cd to adults and children in the crops… - this part of the sentence should be revised.

*Response: Thank you for your kind advice. This sentence will be rephrased in our revised manuscript as follows: "The human health risks of consuming crops were assessed by the Hazard Quotient (HQ) according to the method introduced by Environmental Protection Agency (EPA) in the US.".* **(Materials and method: Line 151-153)**

➢ Line 157. Please define Hazard Quotient – what it indicates actually.

*Response: Thank you for your kind advice. We have clearly explained the Hazard Quotient. Hazard Quotient has been widely used to estimate the health risk related to the special polluted crops with heavy metals according to the method introduced by Environmental Protection Agency (EPA) in the US. Hazard Quotient (HQ) values were calculated as the following formula: HQ = (EF x ED x C x IR)/(BW x AT x RfD) (Wei et al., 2020). For human health, HQ lower than 1 demonstrates no risk (Mehdizadeh et al., 2021).*

*Where EF (Exposure Frequency): 365 days/year.*

*ED (Exposure Duration): 70 years for adult, 7 years for children.*

*C: Cd concentrations in the rice grain and oilseed (mg/kg).*

*IR (Ingestion Rate): For rice grain, 0.3892 kg/day for adult and 0.1984 kg/day for children, respectively. For rape oil, 0.025 kg/day for adult and 0.0125 kg/days for children, respectively.*

*BW (Body Weight): 62.71 kg for adult male, 55.1 kg for adult female and 25.6 kg for children.*

*AT (Averaging Time): 25550 days for adult and 2555 days for children.*

*RfD (Reference of Dose): 0.001 mg/kg for Cd.*

**(Materials and method: Line 153-166)**

*References:*

*Wei, R., Wang, X., Tang, W., Yang, Y., Gao, Y., Zhong, H., Yang, L., 2020. Bioaccumulations and potential human health risks assessment of heavy metals in ppk-expressing transgenic rice. Science of The Total Environment 710, 136496.*

*Mehdizadeh, L., Farsaraei, S., Moghaddam, M., 2021. Biochar application modified growth and physiological parameters of Ocimum ciliatum L. and reduced human risk assessment under cadmium stress. J Hazard Mater 409, 124954.*

➢ Line 158. Please write HQ in full when it is in the beginning of a sentence. Please follow this for other cases throughout the manuscript.

***Response:*** *Thank you for your kind advice. The full name of HQ has been explained at the beginning of the sentence in our revised manuscript. The new sentence is: "Hazard Quotient values were calculated using the following equation". Meanwhile, we carefully checked similar mistakes throughout the manuscript.* **(Materials and method: Line 154)**

➢ Line 159. calculated using the following equation:

***Response:*** *Thank you for your kind advice. This sentence was rephrased as follows: "Hazard Quotient values were calculated using the following equation.".* **(Materials and method: Line 154-155)**

➢ Line 160. (1) can be removed.

***Response:*** *Thank you for your kind advice. The (1) was removed in our revised manuscript.*

➢ Line 161. Please write "where," beforehand.

***Response:*** *Thank you for your kind advice. The word "where" was added in the*

*beginning of this sentence.*    **(Materials and method: Line 157)**

➢ Line 174. What is actually meant by "all statistics"? Should it be graphical interpretation or so…?

***Response:*** *We are sorry for our unclear writing. In this study, Data statistical significance was analyzed using SPSS 18.0 package, and means values were considered to be different when P < 0.05 using least significant difference (LSD). Figures were performed using Origin 8.0 (USA). In our revised manuscript, we have re-written this section.*    **(Materials and method: Line 170)**

Results

➢ Line 175. 'Results' only.

***Response:*** *We are sorry for our carelessness. This mistake has been corrected in our revised manuscript.*    **(Results: Line 171)**

➢ Line 180. OM is generally expressed in %.

***Response:*** *Thank you for your kind advice. We have changed the unit of organic matter.*    **(Results: Line 176-178)**

➢ Line 182. Please rephrase. What is indicated by 'reached' here?

***Response:*** *Thank you for your kind advice. The sentence has been rephrased as follows: "The carbon percentage of biochar was 92.50%.".*    **(Results: Line 178)**

➢ Line 197. Please see the comment made before; compare with the line no. 38….

***Response:*** *We are sorry for our carelessness. The "AcOH-extractable Cd" was replaced to "HOAc-extractable Cd".*    **(Results: Line 190)**

➢ Line 197-198. The second part of the sentence should be in the discussion section. Please use reference(s) for such statement.

*Response:* *Thank you for your kind advice. This sentence has been removed in our revised manuscript.*

➢ Line 200. … flooding irrigation… Please use one.

*Response:* *Thank you for your kind advice. This sentence has been removed in our revised manuscript.*

➢ Line 212-213. Not true. Please rephrase with actual observations from the figure.

*Response:* *we are sorry for our incorrect writing. This sentence has been revised to: "As shown in Figure 3, the application of biochar and oyster shell reduced the Cd contents in brown rice and oilseed".* **(Results: Line 202-203)**

➢ Line 213-214. Please rephrase "treatments without amendments".   Please check the Figure 3a to confirm the value 0.88 mg/kg.

*Response:* *We are sorry for our carelessness. The mean value of brown rice in PA0 was 0.91 mg/kg. This sentence has been revised to: "In the PA0 treatments, the Cd content in brown rice was 0.91 mg/kg."* **(Results: Line 204-205)**

➢ Line 216. … oyster shell had the ….

*Response:* *Thank you for your kind advice. The sentence has been rephrased to: "Compared to control (PA0), the Cd content in brown rice was decreased by 20.88% in PA1 and 30.77% in PA2, indicating that oyster shell had the superior of Cd immobilization capacity than biochar".* **(Results: Line 204-205)**

➢ Line 220. Oilseed Cd content in RT-PA3 was not significantly lower than that in the Control. Please check and amend.

*Response:* *We are sorry for our inaccurate writing. We have carefully checked and revise this sentence as follows: "In addition, the Cd contents in oilseed was reduced in the PA1 and PA3, about 27.63% and 19.74% lower than that in PA0, respectively.".*
 **(Results: Line 206)**

➢ Line 221. "The results indicated that biochar and oyster shell application…" – sole application or combine application?

*Response: We are sorry for our unclear statement. This sentence was revised to be better understanding: "Moreover, the cooperative application of biochar and oyster shell contributed to higher reduction of Cd in brown rice (29.67%) than that in signal biochar (20.88%). ".* **(Results: Line 207)**

➢ Line 224. Please write HQ in full as in the beginning of the sentence.

*Response: Thank you for your kind advice. The sentence has been revised in our revised manuscript.* **(Results: Line 211)**

➢ Line 224-225. This sentence can be removed. Rather, 'Figure 4' can be written at the end of the sentence in line no. 227.

*Response: Thank you for your kind advice. This sentence has been removed in the revised manuscript.*

➢ Line 226. …. adult female > adult male, which indicated that children were….

*Response: Thank you for your kind advice. The sentence has been rephrased in our revised manuscript as follows: "The HQ order of consuming brown rice and oilseed was children > adult female > adult male, which indicated that children had more health risk than adults for the intake of contaminated crops.".* **(Results: Line 213)**

➢ Line 226. Please use a more appropriate word for 'sensitive'.

*Response: Thank you for your kind advice. The sentence has been revised as follows: "The order of HQ for consuming brown rice and oilseed was children > adult female > adult male, which indicated that children had more health risk than adults after the intake of contaminated crops".* **(Results: Line 213)**

➢ Line 228. … HQ values of consuming brown rice for adult male, adult female and children reached 5.46, 6.21 and 6.82, respectively. – What does it mean by

"reached"? Is this the max value for a certain group? Moreover, the values given here do not match with the observations in the Figure 4a. Please check and amend clearly.

*Response: We are sorry for this mistake. The values were the average mean. This sentence has been revised to better understand in our revised manuscript. The new sentence is: "Without the application of amendments, HQ values of consuming brown rice for adult male, adult female and children were 5.66, 6.44 and 7.07, respectively.".* **(Results: Line 215-216)**

➤ Line 229. … HQ values for brown rice intake…

*Response: Thank you for your kind advice. According to your kind advice, this sentence has been revised to: " For children, HQ values for brown rice intake in R-PA1, R-PA2 and R-PA3 were decreased by 20.87%, 31.11% and 29.76%, respectively. ".* **(Results: Line 217)**

➤ Line 229-231. Please rephrase. The decrease in the HQ values for which group(s) – adult male female or children?

*Response: Thank you for your kind advice. According to your kind advice, the sentence has been rephrased in our revised manuscript as follows: "For children, HQ values for brown rice intake in R-PA1, R-PA2 and R-PA3 were decreased by 20.87%, 31.11% and 29.76%, respectively,* compared to control.". **(Results: Line 216)**

➤ Line 231-233. Please rephrase. What does the "significantly" mean here?

*Response: Thank you for your kind advice. As shown in Figure 4. The HQ values of consuming oilseed and brown rice were 0.022 - 0.030 and 3.903 - 7.075, respectively. In our revised manuscript, this sentence has been rephrased to better understanding. The new sentence is: "In addition, it was also observed that the application of amendments decreased the HQ values of consuming oilseed by 17.27 - 28.14% compared to the control."* **(Results: Line 219-220)**

➤ Line 237. It should be Figure 5a, not 2a.

*Response: We are sorry for our mistake. This mistake has corrected in our revised manuscript.* **(Results: Line 224)**

➤ Line 239. Please avoid using 'points' for indicating the pH values.

*Response: Thank you for your kind advice. According to your kind advice, this sentence was revised to: "Meanwhile, the cooperative of biochar and oyster shell also increased soil pH to 7.10 - 7.24."* **(Results: Line 226)**

➤ Line 240. slightly, not slight.

*Response: We are sorry for our carelessness. This mistake has been corrected in our revised manuscript. The new sentence is: "The application of oyster shell slightly increased the CEC of soil in the rice planting, while both oyster shell and biochar had no significant effects on the CEC of soil in the oilseed rape planting (Figure 5b).".* **(Results: Line 227)**

➤ Line 241. CEC of soil….

*Response: Thank you for your kind advice. According to your advice, the sentence has been rephrased in our revised manuscript. The new sentence is: "The application of oyster shell slightly increased the CEC of soil in the rice planting, while it had no significant effects on CEC of soil in the oilseed rape planting (Figure 2b).".* **(Results: Line 228)**

➤ Line 242. Figure 5b, not 2b.

*Response: We are sorry for our carelessness. This mistake has been corrected in our revised manuscript.* **(Results: Line 229)**

➤ Line 244-245. This is irrelevant here.

*Response: Thank you for your kind advice. According to your advice, the irrelevant*

*statement was deleted in our revised manuscript.*

➤ Line 247. Table S2 is missing.

*Response: Thank you for your kind advice. Table S2 was placed in the supplementary materials.* **(Supplementary Materials: Table S2)**

➤ Line 260-262. Please rephrase.

*Response: Thank you for your kind advice. This sentence has been rephrased in our revised manuscript. The new sentence is: "In the rice-oilseed rape rotation, the application of biochar (PA1) increased the dehydrogenase activity, about 20.12 - 25.49% higher than that of control (PA0).".* **(Results: Line 243-244)**

➤ Line 262-264. Please rephrase.

*Response: Thank you for your kind advice. The sentence has been rephrased in our revised manuscript as follows: "Urease activity was markedly enhanced by the oyster shell treatment.".* **(Results: Line 245)**

➤ Line 265-266. Please rephrase.

*Response: Thank you for your kind advice. The sentence has been rephrased in our revised manuscript. The new sentence is: "However, biochar had no obvious effects on the activities of acid phosphate and invertase, but oyster shell significantly reduced the acid phosphate activity by 43.30% in the rice planting.".* **(Results: Line 248-249)**

➤ Line 267. …. biochar had no negative …

*Response: Thank you for your kind advice. According to your advice, the sentence has been rephrased as follows: "In addition, the application of biochar had no negative effects on invertase activity, while oyster shell slightly decreased the invertase activity on rice-oilseed rape rotation.".* **(Results: Line 248-249)**

➤ Line 269-270. Please make it clear whether it was individual application of the treatments or combined application.

*Response:* *Thank you for your kind advice. The sentence has been rephrased to better understanding in our revised manuscript. The new sentence is: "In addition, the cooperative application of biochar and oyster shell enhanced the activities of catalase and β-galactosidase activity by 10.71 - 34.31% and 82.08 - 244.38%, respectively, compared to control.".* **(Results: Line 250-252)**

➤ Line 274. Please rephrase "to experimental data".

*Response:* *Thank you for your kind advice. The sentence has been revised in our revised manuscript. The new sentence is: "The Pearson correlation analysis was used to analyze the relationship among different parameters.".* **(Results: Line 254-255)**

➤ Line 275. Person correlation value is expressed by 'r'. Please check this throughout the manuscript.

*Response:* *We are sorry for this mistake. This mistake was corrected in our revised manuscript.* **(Results: Line 256)**

➤ Line 278. The Figure 7b….

*Response:* *We are sorry for our carelessness. This mistake has been corrected in our revised manuscript.* **(Results: Line 258)**

➤ Line 278. …. Cd uptake by oilseed rape ….

*Response:* *Thank you for your kind advice. This sentence has been revised in our revised manuscript. The new sentence is: "The Figure 7b showed a weak correlation between Cd uptake by oilseed rape and Cd bioavailability.".* **(Results: Line 259)**

➤ Line 282. Acidic

*Response:* *We are sorry for our carelessness. The "acid" has been revised to "acidic".* **(Results: Line 262)**

➤ Line 287. (detailed information see Supplementary Information) – should be

written in an appropriate way. There is no SI attached with the manuscript.

*Response: Thank you for your kind advice. It has been revised in our manuscript. Meanwhile, the supplementary materials were uploaded in the system of Soil (https://soil.copernicus.org/preprints/soil-2021-145/).* **(Results: Line 267)**

➢ Line 288. The dosage of application was 15 t/ha, i.e., 15000 kg/ha. So, please review this amount and also the calculated cost in line numbers 289 & 290.

*Response: Thank you for your kind advice. We have checked the amount and re-calculated the cost in our revised manuscript.* **(Results: Line 268-269)**

➢ Line 289-290. joint

*Response: We are sorry for our carelessness. This mistake has been corrected in our revised manuscript.* **(Results: Line 269)**

➢ Line 291. Please amend the sentence as '… biochar and oyster shell was found to be more effective….'. Again, how this was more effective than the individual applications? The use of oyster shell alone had either similar effects or its effects were not significantly different than the joint application. Moreover, oyster shell is much cheaper than biochar, and thus, compared to the combined use.

*Response: Thank you for your kind advice. This sentence has been rephrased in our revised manuscript. oyster shell showed a better effect on reducing Cd bioavailability than biochar. For Cd immobilization, there was no significant difference between PA2 (15000 kg/ha oyster shell) and PA3 (7500 kg/ha biochar and 7500 kg/ha oyster shell). However, compared to signal biochar treatment, the cooperative application of oyster shell and biochar could significantly increase the Cd immobilization and decrease the remediation cost. Meanwhile, compared to the signal oyster shell treatment, the cooperative application of oyster shell and biochar could markedly enhance the soil biochemical properties, such as the organic matter, available K, and the activities of dehydrogenase, acid phosphatase and $\beta$-galactosidase. Therefore, the cooperative*

*application of biochar and oyster shell might be a practical way to immobilize Cd and enhance the soil biochemical properties. In this study, the advantages of the cooperative application of biochar and oyster shell were discussed in the discussion section.*

➢ Line 292-294. Should be in the discussion section.

***Response:*** *Thank you for your kind advice. The sentence has been replaced to the discussion section.*

**Discussion**

➢ Line 296. …. are the most …

***Response:*** *We are sorry for this grammatical mistake. This sentence was revised to: "Rice and oilseed rape are the most important crops over the globe.".* **(Discussion: Line 273)**

➢ Line 303-304. Please use a reference for this statement.

***Response:*** *Thank you for your kind advice. We have added a reference for this statement. The new sentence is: "In-situ immobilization was an effective pathway to decrease Cd uptake by crops by the application of amendments (Kumpiene et al., 2008).".* **(Discussion: Line 281)**

*Reference:*

*Kumpiene, J., Lagerkvist, A., Maurice, C., 2008. Stabilization of As, Cr, Cu, Pb and Zn in soil using amendments – A review. Waste Management 28, 215-225.*

➢ Line 305. …. have…

***Response:*** *Thank you for your kind advice. This mistake has been corrected in our revised manuscript. The new sentence is: "Previous studies have revealed that biochar had a great potential on the Cd immobilization by surface absorption and co-precipitation (He et al., 2019; Liu et al., 2018).".* **(Discussion: Line 286)**

*References:*

*He, L., Zhong, H., Liu, G., Dai, Z., Brookes, P.C., Xu, J., 2019. Remediation of heavy metal contaminated soils by biochar: Mechanisms, potential risks and applications in China. Environmental Pollution 252, 846-855.*

*Liu, H., Xu, F., Xie, Y., Wang, C., Zhang, A., Li, L., Xu, H., 2018. Effect of modified coconut shell biochar on availability of heavy metals and biochemical characteristics of soil in multiple heavy metals contaminated soil. Science of The Total Environment 645, 702-709.*

➢ Line 306. … had …

*Response:* *Thank you for your kind advice. This mistake has been corrected in our revised manuscript. The new sentence is: "Previous studies have revealed that biochar had a great potential on the Cd immobilization by surface absorption and co-precipitation (He et al., 2019; Liu et al., 2018).".* **(Discussion: Line 286)**

*References:*

*He, L., Zhong, H., Liu, G., Dai, Z., Brookes, P.C., Xu, J., 2019. Remediation of heavy metal contaminated soils by biochar: Mechanisms, potential risks and applications in China. Environmental Pollution 252, 846-855.*

*Liu, H., Xu, F., Xie, Y., Wang, C., Zhang, A., Li, L., Xu, H., 2018. Effect of modified coconut shell biochar on availability of heavy metals and biochemical characteristics of soil in multiple heavy metals contaminated soil. Science of The Total Environment 645, 702-709.*

➢ Line 306. Please give references for the previous studies.

*Response:* *Thank you for your kind advice. We have added references for this statement. The new sentence is: "Previous studies have revealed that biochar had a great potential on the Cd immobilization by surface absorption and co-precipitation (He et al., 2019; Liu et al., 2018).".* **(Discussion: Line 287)**

*References:*

*He, L., Zhong, H., Liu, G., Dai, Z., Brookes, P.C., Xu, J., 2019. Remediation of heavy*

*metal contaminated soils by biochar: Mechanisms, potential risks and applications in China. Environmental Pollution 252, 846-855.*

*Liu, H., Xu, F., Xie, Y., Wang, C., Zhang, A., Li, L., Xu, H., 2018. Effect of modified coconut shell biochar on availability of heavy metals and biochemical characteristics of soil in multiple heavy metals contaminated soil. Science of The Total Environment 645, 702-709.*

➢ Line 307. Please rephrase the sentence.

*Response: Thank you for your kind advice. This incorrect sentence has been deleted in our revised manuscript.*

➢ Line 310. But the reductions in Cd were not statistically significant compared to the control treatment.

*Response: We are sorry for this statement. We revised this sentence to be more precise. Similar to other reports (Jing et al., 2020; Mehdizadeh et al., 2021), the Cd content in brown rice and oilseed was decreased after the application of biochar and oyster shell.* **(Discussion: Line 300-302)**

*References:*

*Jing, F., Chen, C., Chen, X., Liu, W., Wen, X., Hu, S., Yang, Z., Guo, B., Xu, Y., Yu, Q., 2020. Effects of wheat straw derived biochar on cadmium availability in a paddy soil and its accumulation in rice. Environ Pollut 257, 113592.*

*Mehdizadeh, L., Farsaraei, S., Moghaddam, M., 2021. Biochar application modified growth and physiological parameters of Ocimum ciliatum L. and reduced human risk assessment under cadmium stress. J Hazard Mater 409, 124954.*

➢ Line 311-314. Please rephrase.

*Response: Thank you for your kind advice. The sentence has been rephrased in our revised manuscript as follows: "Furthermore, the health risk related to the special polluted crops consumption with Cd has been estimated by HQ, and the decreased HQ values demonstrated that the human health risk of consuming crops was decreased by the application of amendments (Ma et al., 2021a). ".* **(Discussion: Line**

*203-305*)

*References:*

*Ma, L., Liu, Y., Wu, Y., Wang, Q., Sahito, Z.A., Zhou, Q., Huang, L., Li, T., Feng, Y., 2021a. The effects and health risk assessment of cauliflower co-cropping with Sedum alfredii in cadmium contaminated vegetable field. Environmental Pollution 268, 115869.*

➢ Line 315. Please use an appropriate word for "has" here.

*Response:* *Thank you for your kind advice. We has revised this sentence in our manuscript.* (**Discussion: Line 304**)

➢ Line 315-326. Please rephrase with constructive discussion in the context.

*Response:* *Thank you for your kind advice. This part has been carefully rephrased in our revised manuscript.* (**Discussion: Line 284-300**)

➢ The discussions are mostly linked to the pH; however, the pH for oyster shell and combined treatment are not significantly different, and so the effects due to pH as well. How can this strongly support the joint application over oyster shell alone? The authors can also highlight the benefits of using biochar with oyster shell linking with different other factors.

*Response: Thank you for your kind advice. In our revised manuscript, we have clearly stated the benefits of using biochar with oyster shell linking with different factors. In this study, oyster shell showed a better effect on reducing Cd bioavailability than biochar. Compared with signal biochar treatment, the cooperative application of oyster shell and biochar could markedly increase the Cd immobilization efficiency and decrease the remediation cost. Although there was no significant difference between PA2 (15000 kg/ha oyster shell) and PA3 (7500 kg/ha biochar and 7500 kg/ha oyster shell), the cooperative application of oyster shell and biochar had a better effect on enhancing the soil biochemical properties than signal oyster shell, such as the organic matter, available K, and the activities of dehydrogenase, acid phosphate*

*and β-galactosidase. Therefore, the cooperative application of biochar and oyster shell might be a practical way to immobilize Cd and enhance the soil biochemical properties.* **(Discussion: Line 307-340)**

➢ Line 337. Please write 'P' in full in the beginning of the sentence.

*Response: Thank you for your kind advice. The full name will be added at the beginning of the sentence. The new sentence is: "Phosphorus fractions are mainly dependent on soil pH, soil mineralogy and phosphate fertilizer application.".* **(Discussion: Line 308)**

➢ Line 345. Please avoid the use of "PCA".

*Response: Thank you for your kind advice. We revised this sentence as follows: "Correlation analysis (Figure 7) further demonstrated that available P was highly correlated to the changes of soil pH (r > 0.99).".* **(Discussion: Line 316)**

➢ Line 350-352. Please rephrase.

*Response: Thank you for your kind advice. This sentence has been rephrased in our revised manuscript. The new sentence is: "In this study, the activities of dehydrogenase, urease, catalase and β-galactosidase were increased in the treatments of biochar and oyster shell (Figure 6). ".* **(Discussion: Line 321-322)**

➢ Line 354. Please use appropriate words for 'obvious' and 'stimulation'.

*Response: Thank you for your kind advice. We will use appropriate words to replace to these words. The new sentence is: "In this study, the activities of dehydrogenase, urease, catalase and β-galactosidase were increased in the treatments of biochar and oyster shell (Figure 6). ".* **(Discussion: Line 321-322)**

➢ Line 354-355. Please either remove this sentence or rephrase.

*Response: Thank you for your kind advice. This sentence has been removed in our*

*revised manuscript.*

➢ Line 357. Please add year for the reference used, and remove the reference written at the end of the sentence in the line no. 358.

*Response: Thank you for your kind advice. The incorrect reference format was revised in our manuscript.* **(Discussion: Line 328)**

Conclusions

➢ Lines 372. Please use a more appropriate for 'extraordinary' here?

*Response: Thank you for your kind advice. The "extraordinary" was revised to "great" in our manuscript. The new sentence is: "The application of oyster shell significantly ($p < 0.05$) increased soil pH and thus decreased the bioavailability of Cd in soil."*. **(Conclusions: Line 345)**

➢ Line 378. …. our, not Our.

*Response: We are sorry for our carelessness. This mistake was corrected in our revised manuscript.* **(Conclusions: Line 351)**

**Others**

➢ Figure 7. The values on the figures are not clear.

*Response: Thank you for your kind advice. We has revised the figures to be clear.* **(Figure 7)**

**References**

Please follow the journal style of referencing in the reference section. All the references need to be corrected following the journal style. So, please check and amend throughout the reference section.

➢ **Response:** *Thank you for your kind advice. We have carefully revised the format of references according to the style of this journal.* **(References)**

---

## Author Response (AR2)

**Response to the editor's comments**

**Soil**

**Manuscript No.:** SOIL-2021-145

**Manuscript title:** The application of biochar and oyster shell reduced cadmium uptake by crops and modified soil fertility and enzyme activities in contaminated soil

**Article type:** Research paper

**Authors:** Bin Wu, Jia Li, Mingping Sheng, He Peng, Dinghua Peng, Heng Xu

Comment 1: Please, replace the term "soil biochemical properties" with "main soil enzyme activities" or a similar more acurate term in the title of the manuscript and throughout the text.

Response: Dear editor, we would like to thank you for your kind comments and giving us an opportunity to revise our manuscript. In this study, the soil fertility (organic matter, available P, available N and available K) and main soil enzyme activities (dehydrogenase, acid phosphate activity, urease, catalase, β-galactosidase and invertase) were determined to evaluate the effects of biochar and oyster shell on soil quality. Therefore, we replaced the term "soil biochemical properties" with "soil fertility and enzyme activities" in the title of the manuscript and throughout the text. **(Revised with red color)**

Comment 2: Replace the term "content" with "concentration" in the manuscript (including text, graphs and figure legends).

Response: Thank you for your kind comments. The term "content" was replaced with "concentration" in our revised manuscript (including the text, graphs and figure legends). **(Revised with red color)**